Review

ecology, theoretical biology, evolution

senescence, ageing, life history, trade-off, demography

**Author for correspondence:**
Mark Roper
e-mail: mark.roper@keble.ox.ac.uk

# Senescence: why and where selection gradients might not decline with age

Mark Roper[1], Pol Capdevila[1,4] and Roberto Salguero-Gómez[1,2,3]

[1]Department of Zoology, University of Oxford, 11a Mansfield Road, Oxford OX1 3SZ, UK
[2]Centre for Biodiversity and Conservation Science, University of Queensland, St Lucia, Queensland 4071, Australia
[3]Evolutionary Demography Laboratory, Max Plank Institute for Demographic Research, Rostock 18057, Germany
[4]School of Biological Sciences, University of Bristol, 24 Tyndall Avenue, Bristol BS8 1TQ, UK

MR, 0000-0002-1712-3344

Patterns of ageing across the tree of life are much more diverse than previously thought. Yet, we still do not adequately understand how, why and where across the tree of life a particular pattern of ageing will evolve. An ability to predict ageing patterns requires a firmer understanding of how and why different ecological and evolutionary factors alter the sensitivity of fitness to age-related changes in mortality and reproduction. From this understanding, we can ask why and where selection gradients might not decline with age. Here, we begin by summarizing the recent breadth of literature that is unearthing, empirically and theoretically, the mechanisms that drive variation in patters of senescence. We focus on the relevance of two key parameters, population structure and reproductive value, as key to understanding selection gradients, and therefore senescence. We discuss how growth form, individual trade-offs, stage structure and social interactions may all facilitate differing distributions of these two key parameters than those predicted by classical theory. We argue that these four key aspects can help us understand why patterns of negligible and negative senescence can actually be explained under the same evolutionary framework as classical senescence.

## 1. Classical evolutionary framework of senescence

Senescence, the increasing risk of mortality and decline in reproduction with age after maturity, has long been explained by a collation of theories defining the 'classical evolutionary framework of senescence'. The central logic common to these theories argues that the force of natural selection weakens with age [1–4]. Selection becomes too weak to oppose the accumulation of genes that negatively affect older age classes [1], or favours these genes if they also have beneficial effects at earlier ages in life [2], when the contribution individuals make to future populations is assumed to be greater. Selection should therefore favour resource investment into earlier reproduction rather than late-life maintenance [4], or at least only invest in non-reproductive functions for as long as they would naturally be required in the wild.

Underpinning all evolutionary theories of senescence is the omnipresence of extrinsic mortality. The later age classes of such a cohort should, all else being equal, contribute less to the ancestry of future generations [5]. Medawar [1] predicted this will create a 'selection shadow', whereby late-age-acting deleterious alleles will be less effectively removed by selection (mutation accumulation). Building on this, Williams [2] instead developed the 'antagonistic pleiotropy' theory of ageing [6,7], where he argued that such late-acting detrimental alleles could actually invade under positive selection if they had beneficial pleiotropic effects at younger ages.

Kirkwood [4] also took an optimality approach but framed his 'disposable soma' theory from a physiological perspective. Because it is the germ line that survives through generations, natural selection will only favour a level of somatic investment that optimizes the success of the germline [8]. As laid out in a thought experiment by Partridge & Barton [9], however, a hypothetical

organism composed solely of germ line cells will still experience a non-zero probability of death and/or become unable to reproduce. The intensity of selection on the organism's survival and reproduction will therefore still decline with age, and senescence will still evolve. Evidence that senescence can occur in asexual metazoans [10] and bacteria with asymmetrical division [11] support this notion that a germ–soma divide is neither necessary nor sufficient for the evolution of senescence. Instead, the ultimate requirement for senescence is asymmetry between parent and offspring, essentially age structure, that makes individuals different in the eyes of selection [9].

These verbal ideas for the evolution of senescence were formalized mathematically by Hamilton [3], who used Fisher's [5] assumption that the Malthusian parameter $r$ is an appropriate measure of fitness

$$\int_0^\infty e^{-rx} l(x) m(x) \mathrm{d}x = 1. \tag{1.1}$$

The above equation is the Euler–Lotka equation [7] and $r$, also known as the population growth rate, is the single real root. The terms $l(x)$ and $m(x)$ define the probability of survival to, and reproduction at, age $x$ respectively. Using implicit partial differentiation of equation (1.1), Hamilton calculated the effects on $r$ of small changes in survival or reproduction at some age $x$,

$$\frac{\partial r}{\partial m(x)} \approx \frac{e^{-rx} l(x)}{T} \tag{1.2}$$

and

$$\frac{\partial r}{\partial \mu(x)} \approx - \frac{\int_x^\infty e^{-ry} l(y) m(y) \mathrm{d}y}{T}, \tag{1.3}$$

where $\mu(x) = -(\mathrm{d}\ln l(x)/\mathrm{d}x)$ and defines mortality risk at age $x$, and $T$ is generation time. The resulting quantities represent 'forces of selection' on age-specific vital rates [3]. The larger the absolute value of the quantity, the stronger the response of selection to a given change [7,12]. Equation (1.2) shows how the force of selection on reproduction at age $x$ is proportional to survival at age $x$ should decrease with age in a stationary or growing population, whereas equation (1.3) formulates how the force of selection acting against (hence the negative sign) an increase in age-specific mortality declines from the age at first reproduction [3,7,12–16] (but see Baudisch [17] for other indicators of selection where the forces of selection do not necessarily decline with age).

How extrinsic mortality may drive variation in rates of senescence has been debated off the back of one of Williams's nine predictions (see Gaillard & Lemaître [18] for a review of each) for the evolution of senescence [2]. Williams stated that 'Low adult death rates should be associated with low rates of senescence, and high adult death rates with high rates of senescence' [2]. Although the legitimacy of this prediction is debated [19–21], a general consensus seems to suggest that a change in extrinsic mortality has to be age-dependent to alter the rate of senescence [22]. An age-independent change in mortality, by definition of being age-independent, will mean the overall selection gradient should still follow the same pattern over all age classes if exponential growth is assumed [14,22]. If extrinsic mortality interacts with physiological condition, however, this could alter ageing rates [23]. Williams & Day found such an interaction to have a general tendency to strengthen

selection against senescence at all ages, but potentially stronger effects at younger ages, favouring slower senescence earlier in life and more rapid senescence later in life [23].

To summarize the classical evolutionary framework and subsequent theoretical elaborations for age-structured populations, senescence (i) has not evolved adaptively without pleiotropy, (ii) occurs as a by-product of weak selection, and (iii) requires asymmetry between parent and offspring (i.e. requires age structure). Based on Hamilton's model [3] (equations (1.2) and (1.3)), we should expect to see the risk of mortality rise and reproduction decline throughout adulthood in any age-structured population. As we will show for 475 species of animals and plants, supporting previous research [24], many species in fact display negligible [25] or even negative [26] senescence, where the risk of mortality remains constant or decreases with age, and reproduction remains constant or increases with age. This phenomenon occurs because these species are not solely age-structured, in the sense that chronological age *per se* is not the principal driver of their demography [27]. For such species, selection gradients may not monotonically decline with age throughout adulthood, or a decline may be delayed [28,29].

## 2. Current data

With ever-growing amounts of readily available longitudinal demographic datasets [30,31], comparative demography offers a tool to reveal the diversity in ageing patterns across the tree of life [24,32]. Using high-resolution demographic information for wild populations of 80 animal and 395 plant species worldwide (see electronic supplementary material for methods), we provide a quantitative evaluation of the rates of actuarial senescence—the change in mortality risk with age after maturation—across a subset of multicellular organisms. We use a 'pace-shape' framework of ageing [33,34]. In it, *pace* of ageing quantifies the speed of life via mean life expectancy [34], whereas the *shape* of ageing (i.e. senescence rate) quantifies the spread and timing of mortality events, normalized by mean life expectancy. The shape metric, $S$, is bound between −0.5 and 0.5, where $S > 0$ indicates that more mortality events occur at advanced ages (i.e. positive actuarial senescence), while $S < 0$ indicates low mortality late in life (i.e. negative actuarial senescence [35]). We quantify the role of evolutionary history on actuarial senescence across our 475 species by estimating its phylogenetic signal [36] using phylogenies for animals [37] and plants [38], respectively. Finally, we use derived life tables [39] from both age and stage-based models (see electronic supplementary material) to quantify age-specific reproduction rates ($m(x)$) to evaluate whether they match their patterns of actuarial senescence.

The majority of animal species (59 out of 80 species) display a negligible change in their risk of mortality with age. Positive actuarial senescence is especially scarce across invertebrates in our data, with the water flea (*Daphnia pulex*; figure 1*a*) as the sole example of positive actuarial senescence. The remaining 14 invertebrate species display negligible actuarial senescence, as in the case of the long-wristed hermit crab (*Pagurus longicarpus*; figure 1*a*), or even negative actuarial senescence, for example, the sea whip (*Leptogorgia virgulata*; figure 1*a*). Across vertebrates, 72% of species display little change in the risk of mortality with age (figure 1*a*; electronic supplementary material, table S1). Positively senescent species, however, are

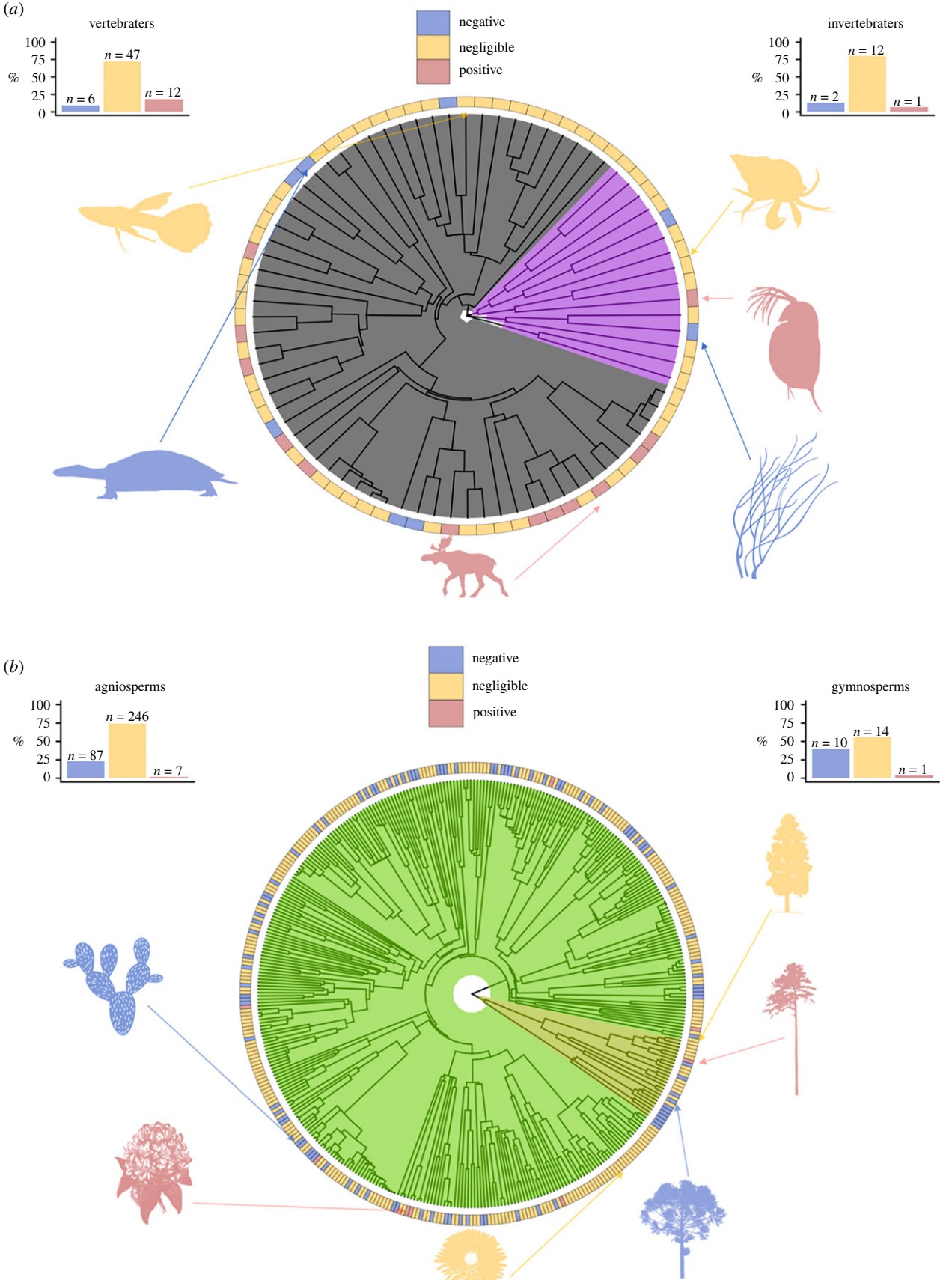

**Figure 1.** The evolution of and escape from senescence across multicellular life. Positive, negligible and negative patterns of actuarial senescence are dispersed throughout the four examined clades, with the percentages of each pattern within each clade shown in the bar charts of each of the figures. (*a*) Actuarial senescence across animals. Depicted around the phylogeny are six representative species, displaying positive (red), negligible (yellow) and negative (blue) senescence from each clade. Clockwise, representing invertebrates, these species are *Pa. longicarpus*, *D. pulex* and *L. virgulata*. For vertebrates, again clockwise, these species are *A. alces*, *P. expansa* and *Poecilia reticulata*. (*b*) Actuarial senescence across plants. Depicted around the phylogeny are six representative species, displaying positive (red), negligible (yellow) and negative (blue) senescence from each clade. For gymnosperms, these species are *Pi. lambertiana*, *Pi. sylvestris* and *Taxus floridana*. For angiosperms, these species are *Hypochaeris radicata*, *Rhododendron maximum* and *Opuntia rastrera*. (Online version in colour.)

Proc. R. Soc. B 288: 20210851

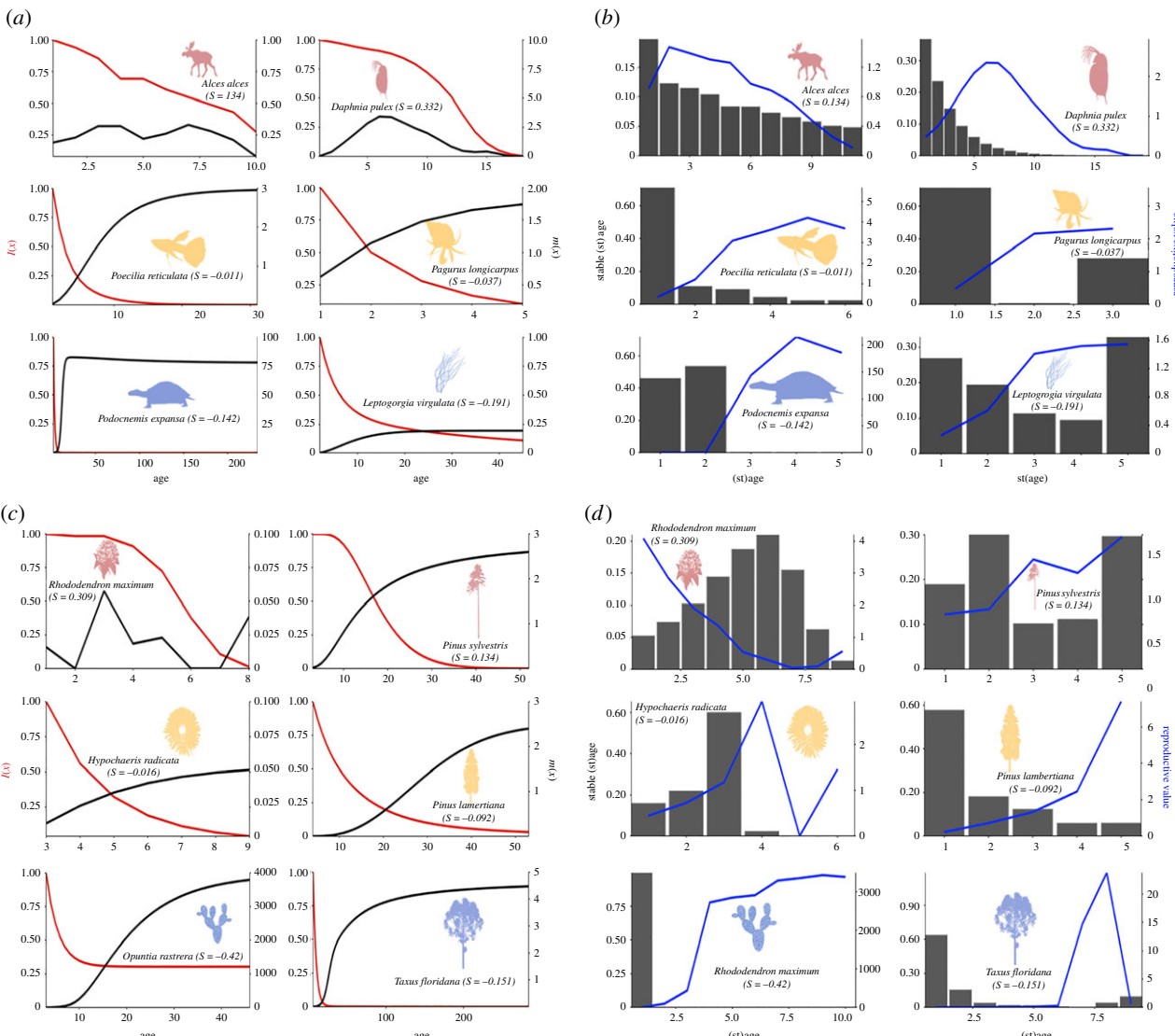

**Figure 2.** Age-based patterns of survivorship ($l(x)$—red) and reproduction ($m(x)$—black) are often decoupled, whereas reproductive value and stable age distributions predict the patterns of senescence. (a) $l(x)$ and $m(x)$ trajectories for the six selected animal species from figure 1. (b) Stable (st)age and reproductive value trajectories for the six selected animal species from figure 1. (c) $l(x)$ and $m(x)$ trajectories for the six selected plant species from figure 1. (d) Stable (st)age and reproductive value trajectories for the six selected plant species from figure 1. $l(x)$ and $m(x)$ trajectories are conditional upon reaching the age of maturity, at which the mature cohort is defined to have entered adulthood with a survivorship of 1. The trajectories of $l(x)$ and $m(x)$ run from age at maturity to the age at which 5% of the mature cohort is still alive. Stable (st)age and reproductive value trajectories are displayed for the whole life course. (Online version in colour.)

more common in vertebrates (18%; 12 out of 65 species) than invertebrates (6%; 1 out of 15 species); these species are primarily mammals (75%; electronic supplementary material, table S1) such as the moose (*Alces alces*; figure 1a). The majority of examined plant species also display negligible senescence. Indeed, only 2% of the 375 examined plant species exhibit positive senescence, including the scots pine (*Pinus sylvestris*) and the great laurel (*Rhododendron maximum*; figure 1b). Approximately 23% of angiosperms show a decreasing risk of mortality with age (e.g. *Opuntia rastrera*; figure 1b), compared to 40% of gymnosperm species (e.g. *Pinus lambertiana*; figure 1b). Overall, 98% of our studied plant species do not display a substantial increasing risk of mortality with age.

Estimates of phylogenetic signal on actuarial senescence are weak across the pool of examined animals (electronic supplementary material, figure S1 and table S3). Specifically, Pagel's $\lambda$ [36], which ranges from zero (weak signal) to one (strong signal), is not significantly different from zero for across all the examined animal species ($\lambda = 0.22$, $p = 0.18$),

nor when considering vertebrates and invertebrates separately (electronic supplementary material, table S3). These results indicate that the patterns of senescence across animals cannot be explained by phylogenetic relatedness, assuming a Brownian model of evolution. More thorough phylogenetic analyses are required, however, to rule out the effects of ancestral constraint on the patterns of senescence in animals. On the other hand, phylogenetic relatedness plays some role in senescence patterns across plants (electronic supplementary material, figure S2 and table S3). The analysis including angiosperms and gymnosperms raises a Pagel's $\lambda$ of 0.31 ($p < 0.001$), most likely due to the significant phylogenetic signal in angiosperms ($\lambda = 0.27$, $p = 0.001$), compared to the non-significant signal in gymnosperms ($\lambda = 0.27$, $p = 0.08$).

Patterns of reproduction ($m(x)$) are diverse and appear somewhat independent of whether the examined species displays actuarial senescence (figure 2a,c; electronic supplementary material, figure S3). In plants, for example, both the scots pine and the great laurel display actuarial

senescence (figure 1b), but their reproductive outputs do not decline with age (figure 2c). This pattern is contrasting to both examples of animals displaying positive senescence, where the moose (A. alces) and water flea (Daphnia pulex) also display reproductive decline with age (figure 2a). The flatweed exhibits negligible actuarial senescence and a relatively constant $m(x)$ trajectory, whereas the long-wristed hermit crab and the sugar pine also display negligible actuarial senescence but have increasing $m(x)$ trajectories. It appears from our study species that survivorship and reproduction can follow independent age-based trajectories.

Long-term individually based studies remain the gold standard for studies of senescence [40–42]. Only by tracking cohorts of individuals throughout their entire lifespans—or by being able to back-track age linked to performance [43]—can one account for variation in quality of individuals, and the subsequent issues of selective appearance and disappearance [44]. Long-term individually based studies are currently taxonomically biased [45], however, mostly limited to birds and mammals [46]. The matrix population models we use here, and the robust methods to derive age-based trajectories of survival and reproduction from them [27], offer an insight into the diversity of ageing rates in a taxonomically broad manner. By observing these broad-scale patterns, we can now isolate and understand these mechanisms behind this variation of age trajectories of mortality and reproduction, elucidating why some species succumb to senescence while others appear to escape its forces [28,29].

## 3. Why and where species may not senesce

It is practical to refer back to Hamilton's models as a starting point. Caswell's [15] reformulation of equations (1.2) and (1.3) show what factors drive the change in the force of selection with age [14–16],

$$\frac{\partial r}{\partial m(x)} \approx \frac{c(x)}{bT} \tag{3.1}$$

and

$$\frac{\partial r}{\partial \mu(x)} \approx \frac{-c(x)v(x)}{bT}, \tag{3.2}$$

where $c(x)$ gives the proportion of individuals aged $x$ at the stable age distribution, $v(x)$ is the reproductive value [5] (the average contribution to the ancestry of future generations), $b$ is the birth rate $\left(b = \left[\int_0^\infty e^{-rx}l(x)\,dx\right]^{-1}\right)$ and $bT$ scales the values of the force of selection. For any given (st)age of a life cycle, the force of selection on an increase in the reproduction or mortality of that (st)age is proportional to the product of two key components: (i) the stable age distribution of individuals at that (st)age and (ii) the reproductive value of individuals that the (st)age contributes to the population (new offspring or their surviving selves). In equation (3.1), the stable age distribution of individuals aged $x$ is weighted by the reproductive value of newborns, which is defined as 1. Importantly here, selection gradients need not always decline with age, or their decline may be delayed. This is because demographic or ecological factors may alter the stable age distribution or reproductive value profile of a population in such a way that maintains stronger selection with age.

Figure 2b,d exemplifies the importance of reproductive value and population structure for explaining divergent patterns of senescence (see also electronic supplementary material, figure S3). All species (except Pinus sylvestris; figure 2d) display reproductive value and stable age distributions that match their classification of actuarial senescence. Species that display positive actuarial senescence have distributions biased to earlier (st)ages, and those that do not have distributions that are more heavily skewed towards later (st)ages. Our assumption of each population sampled being at a stable state at the time of data collection provides the most intuitive reason why the patterns (including for Pinus sylvestris) may not match. It is clear from our analysis that reproductive value and population structure are key to understanding variation in patterns of senescence.

In identifying ecological and evolutionary mechanisms that might lead to alternative ageing patterns than those suggested by Hamilton, we suggest to ask two questions:

(1) How does the ecological and/or evolutionary mechanism alter the reproductive value and stable age distributions of the population from that expected of a standard age-structured population?
(2) How do the resulting distributions affect the evolutionary outcome for senescence?

Here, we consider how (i) the many components of reproductive senescence and trade-offs between survival and reproduction, (ii) the stage structuring of populations, and finally (iii) the spatial structuring of populations can act as ecological and evolutionary mechanisms that alter reproductive value and stable age distributions in ways that drive the evolution of senescence in different directions than those predicted from the classical evolutionary framework.

### (a) Reproductive senescence

Survival and reproduction are intrinsically linked; the evolutionary outcomes for one make little sense without consideration of the other. Considered in isolation, however, there is a relative simplicity afforded by the monotonic nature of survival curves—everybody must eventually die. This simplicity has facilitated the conception of measures of shape [34], entropy [33] or lifespan equality [47] to describe the distribution of deaths, and afforded the ability to describe mortality curves with simple functions (e.g. Weibull or Gomerptz models [48,49]). Only recently have metrics become available to also quantify rates of reproductive senescence across species [35,50]. Unlike survivorship, reproductive schedules can be highly variable, including multiple peaks. This greater complexity in reproduction has made extensive cross-species comparisons more challenging for reproductive senescence.

As reviewed by Lemaître & Gaillard [51], quantifying reproductive senescence is complicated by the multitude of different components that can comprise the reproductive schedule. When broken down into these components, from egg production to juvenile survival and everything in between, heterogeneous rates of reproductive senescence can be revealed. In the grey-leaved cistus (Cistus albidus), for example, older plants produce fewer flowers but show no evidence of a decline in germination capacity [52]. In meerkats (Suricata suricatta), litter size displays a concave quadratic relationship with maternal age, but litter survival is independent of maternal age [53]. Parental effect senescence, where offspring age-specific vital rates are affected by the age of their parents

[54], is also rife but variable across species [55]. Accounting for these heterogenous rates and trade-offs between these different components of the reproductive schedule will be of paramount importance for future studies of reproductive senescence [51].

Adding further complexity to quantifying reproductive senescence is the variety of shapes age-specific reproductive trajectories can take. Across birds and mammals, increases in a component of reproduction followed by decline from a peak are common [56–58], but indefinite increases with age can also occur across taxa [24] (electronic supplementary material, figure S3). In plants, vegetative dormancy [59] and seed banks [60] add remarkable extra layers of complexity [61,62]. Baudisch & Stott's [35] pace and shape metrics of fertility offer an attempt to condense this complexity into two metrics using the insight of cumulative reproduction. Our results here show that age trajectories of reproduction are often uncoupled from the pattern of actuarial senescence (figure 2; electronic supplementary material, figure S3). Species-specific studies that investigate both actuarial and reproductive senescence are becoming more frequent, and both uncoupling [63] and concurrent [64] patterns of actuarial and reproductive senescence appear common.

Overall, the prediction from Hamilton's model for actuarial senescence is clear: senescence should occur in any age-structured population, and start at the age of first reproduction [12]. Equation (3.1) tells us that the force of selection for reproduction is proportional to the abundance of mothers at a given age [14]. With efflux of the oldest due to mortality and influx of the youngest due to births, overall stable age distributions are likely to often be biased towards younger age classes [14]. Yet, there is no theoretical expectation that the stable age distribution will be centered at the age of maturity. Future theoretical work should seek to pinpoint how ecological and demographic forces will shape the stable age distribution, allowing for clearer hypothesized comparative tests for the onset, rate and/or escape from reproductive senescence. Links can then be made to how reproductive and survivorship trajectories should co-evolve when constrained by physiological trade-offs [65,66], unveiling mechanisms behind the variation in synchrony of actuarial and reproductive senescence across species. Only by studying age-patterns of reproduction and mortality in tandem, can we fully understand how different ecological and demographic mechanisms have promoted different patterns of senescence.

## (b) Stage structure and growth forms

Not all species' mortality and reproduction patterns are best predicted by age. Caswell & Salguero-Gómez [67] showed how within a stage, age-specific selection gradients can actually increase with age, a pattern that might commonly be found in plants [32]. Spanning across taxa, many stage-categorized species have the capacity to grow indefinitely [26,68]. Extremely common in plants [69], indeterminate growth is also found in insects [70], fish [71], reptiles [72] and corals [73]. A pioneering study by Vaupel *et al.* [26] hypothesized that such indeterminate growth can favour negative senescence. The authors modelled an organism whose reproductive capacity increases with size. For such an organism, it can pay to sacrifice current reproductive output if such a sacrifice markedly increases size, and, therefore, potential future reproductive output. An increase in size over time implicitly carries with it an increase in age. Negative senescence, a decrease in mortality rate from the age at reproduction and/or increasing reproduction with age, may, therefore, be observed.

In our display of currently available demographic data, 98% of studied plant species and all of our studied corals show little evidence of an increase in risk of mortality with age (e.g. *Paramuricea clavata*; figure 1; electronic supplementary material, tables S1 and S2). We also found evidence for negative actuarial senescence in the South American river turtle (*Podocnemis expansa*; figure 1), an indeterminately growing reptile. Vaupel and colleagues suggested that negligible and negative patterns of senescence may mostly be characteristic of species that attain sizes at reproductive maturity that are not their maximum size, and gain significant reproductive capacity as they grow. Some indeterminately growing species do still display senescence [74], and perhaps the current treatment of growth form as a discrete variable (determinate versus indeterminate) offers limited scope for predictions [28,75]. Transforming growth form into a continuous variable, for example, as size at maturity as a proportion of maximum size, will allow for more quantitative comparative tests on the effects of growth form on senescence.

The concept of negative senescence can be generalized as a resource allocation decision based on a trade-off between current and future reproduction within a life course [76]. Often, this trade-off favours the former [1–4], but Vaupel *et al.*'s insight is that growth can provide a mechanism shifting the optimal balance of the trade-off in the favour of future reproduction. If reproduction is much lower at younger, mature ages (lower sizes) and increases disproportionately with size and age, then the relative reproductive value of older ages classes and the abundance of mothers will be biased towards older age classes. Growth and increasing reproductive capacity with age provide just one mechanism to alter a population's reproductive value and stable age distributions in favour of delaying senescence.

## (c) Social interactions

Hamilton's model assumed an infinite population in which individuals do not interact. A recent themed issue in *Philosophical Transactions B* 'Ageing and sociality: why, when and how does sociality change ageing patterns?' [77] is testament to a growing interest that such interactions between individuals could alter the evolution of senescence [78]. Given the strong relationship between the social environment and mortality risk in humans [79], it is perhaps not surprising that researchers have now begun to focus their attention on trying to test for similar phenomena in other social animals.

Sociality appears to be correlated to some extent with longer lifespan [58,80,81]. Extreme sociality is associated with a 100-fold increase in lifespan in social insects [80]; cooperative breeding birds live longer on average than non-cooperative birds [81] and a recent comparative review of reproductive senescence in birds and mammals reveals a relationship between proxies of sociality and a slower pace of life [58]. However, current evidence suggests that this relationship is likely to be due to longer lifespan first being driven by a reduced threat of extrinsic mortality [81]. Lower individual turnover and overlapping generations then facilitate more opportunities for individuals to interact, and can favour the evolution of cooperation [82,83]. Quantitative theoretical predictions are lacking as to how, or even if, more complex sociality can causally alter selection on age-specific mortality and reproduction, and, therefore, senescence.

Some theoretical treatments for senescence in a social context do exist, with the most influential being that of Lee [84]. Lee extended Hamilton's models to include a 'transfer effect', whereby the author showed how selection gradients on mortality and reproduction can also depend on resources given and received throughout the life course of the individual. Most other models have focused specifically on the interaction between limited dispersal and senescence. These studies have shown that actuarial senescence may favour reduced juvenile dispersal with increasing maternal age [85], that reduced dispersal may favour the evolution of shorter lifespans [86], and that in some scenarios, actuarial senescence can actually be favoured in viscous populations [87,88]. In general, however, a comprehensive ageing theory of social animals is lacking [77].

Cooperative breeding systems, defined by the presence of alloparental care by helpers [89], offer an excellent system for empirical and theoretical study for how social interactions could directly alter patterns of senescence. For example, helpers can provide load-lightening benefits to breeders [80] that may delay breeder senescence and extend lifespan [90–92]. Helping tendencies may also change with age [93], due, for example, to changing relatedness to its group as an individual helper ages [94], or to changing benefits of independent reproduction [95]. Age trajectories of how individuals could affect both helper and breeder senescence profiles, and lead to plastic senescence patterns within a species.

The classification of helpers and breeders, or subordinates and dominants, means that cooperative breeding systems can be modelled as stage-structured populations. Rank or stage may be quantified by the proportional access of total group reproduction an individual gains, which will be determined by the level of reproductive skew within the group [96]. In more despotic societies, such as the naked mole-rat [97] (*Heterocephalus glaber*), the breeding pair of individuals control all reproduction, whereas in plural breeding societies such as Guira cuckoos [98] (*Guira guira*), all individuals have access to reproductive opportunities. This stage-related access to reproduction will bias the reproductive value distribution towards certain classes of individuals, which if they correlate with age, will alter late-life selection.

Furthermore, groups can be composed of kin, non-kin or a mix, thus also potentially creating indirect fitness trajectories of survival or reproduction [99]. The formation of these different types of groups via dispersal patterns will drive the competitive dynamics within groups [100], and how the relatedness of an individual to their group changes with age, which can have extreme consequences such as favouring complete reproductive suppression [94]. The interactions of all of these features of cooperative breeding systems (e.g. helping, stage structure, and compeititve and cooperative interactions) may lead to novel predictions for the evolution of senescence when individuals interact with one another.

# 4. Extending the classical evolutionary framework of senescence

The decline in the force of natural selection with age provides an ultimate explanation for the counterintuitive prevalance of senescence in nature [3]. We have reviewed recent—and provided new—reasons why this decline in the force of natural selection may be postponed, or even reversed during adulthood. Although the forces of selection may always ultimately decline in the extremes of any species's lifespan [26], the variable patterns of ageing from maturity for both survival and reproduction (figures 1 and 2; electronic supplementary material, figure S3) observed in nature [24] deserve explanation [28,29].

We have found it conceptually useful to consider lifespan and senescence as orthogonal characterisitcs of a species, using a pace-shape framework [34]. Indeed, there is now strong evidence that senescence rates are independent of a species's lifespan [24,101]. Discussions often conflate the two, which can be confusing for predictions and prevent further progress in the field. For example, a delayed age of maturity will be associated with longer lifespan and postponed onset of senescence, but should not *a priori* alter the rate of senescence. Likewise, an age (and condition [23]) independent reduced threat of extrinsic mortality should select for longer lifespan but not, all else being equal, make any difference to the pattern of senescence [22]. A consensus on a definition of senescence as the change in mortality and reproduction with age independent of lifespan, we argue, is the first important, albeit trivial, step towards extending the framework.

The identification of different mechanisms that might be associated with variable ageing patterns has relied upon research of previously underexplored taxa. High-resolution demographic data on some major groups of species are still yet limited [102]. We hope that our work contributes to stimulating the next generation of demographers to explore these under-represented corners of biodiversity. There are also other potential mechanisms that may shape senescence outcomes not explored here, such as how sexual selection interacts with ageing rates [103]. It may be easy for one to then imagine any number of mechanisms that could alter ageing patterns, so how can we provide a rigorous framework for accepting or declining these mechanisms?

We show in our review of theory and analyses (figure 2b,d; electronic supplementary material, figure S3) that reproductive value and population structure are key to understanding senescence. The two-question process outlined in §3 can offer a stepping stone framework for evolutionary research of senescence. Senescence can be viewed as a result of early versus late-life trade-offs [104]. Stage structure, whether by size, social hierarchies or division of labour [105], provide mechanisms to weight this trade-off in favour of late life by shifting the population's distributions of reproductive value and class frequencies to reduce the decline of selection in later age classes. Ultimately, as Hamilton displayed [3], and Caswell made explicit [15], it is these two fundamental properties of a population that determine the strength of selection, which in turn determines the ease at which late-acting detrimental alleles may be able to invade. We must first ask how a mechanism can shift these distributions, which will then allow us understand which patterns of senescence will evolve. Hamilton's work provided fundamental insights for the evolution of senescence. Now that we have access to an unparalleled amount of data from a more diverse range of species, it is our duty as demographers, empiricists and theoreticians, to extend the framework and once and for all solve the problem of the evolution of senescence.

Data accessibility. Data are available from the COMPADRE Plant Matrix Database and COMADRE Animal Matrix Database (www.compadre-db.com). Code used for analysis is available in the electronic supplementary material.

The data are provided in electronic supplementary material [106].

**Authors' contributions.** M.R. and R.S.-G. conceived the review project. M.R. and P.C. conducted the analyses with input from R.S.-G. and produced all visualizations. M.R. drafted the first version and, together with P.C. and R.S.-G., revised and edited the manuscript.

All authors gave final approval for publication and agreed to be held accountable for the work performed therein.

**Competing interests.** The authors declare no competing interests.

**Funding.** M.R. was supported by funding from the Biotechnology and Biological Sciences Research Council (BBSRC) (grant no. BB/M011224/1). P.C. was supported by a Ramón Areces Foundation Postdoctoral Scholarship. This research emerged through funding by NERC (NE/M018458/1) to R.S.-G.

**Acknowledgements.** We thank P. Barks, I. Stott and M. Bonsall for key input in the analyses of actuarial senescence. We also thank three anonymous reviewers for comments which greatly improved the manuscript.

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
