## [Peer Review File · Proceedings of the Royal Society B: Biological Sciences]

Review History

RSPB-2021-0851.R0 (Original submission)

Review form: Reviewer 1

Recommendation

Accept with minor revision (please list in comments)

Scientific importance: Is the manuscript an original and important contribution to its field?

Excellent

General interest: Is the paper of sufficient general interest?

Excellent

Quality of the paper: Is the overall quality of the paper suitable?

Excellent

Is the length of the paper justified?

Yes

Should the paper be seen by a specialist statistical reviewer?

Yes

Do you have any concerns about statistical analyses in this paper? If so, please specify them explicitly in your report.

Yes

It is a condition of publication that authors make their supporting data, code and materials available - either as supplementary material or hosted in an external repository. Please rate, if applicable, the supporting data on the following criteria.

Is it accessible?

Yes

Is it clear?

Yes

Is it adequate?

Yes

Do you have any ethical concerns with this paper?

No

Comments to the Author

The manuscript by Roper et al. titled:

“Senescence across the furthest reaches of the universe? Why and where selection gradients might not decline with age.” is beautifully written, an absolute pleasure to read.

The authors first provide an excellent review of the senescence literature and provide an overview of the empirical and theoretical mechanisms that have been proposed to drive the variability of senescence patterns. Then they offer a framework to support their theory, that only by deciphering the underlying factors of age-related changes in mortality and reproduction can one understand why aging has evolved across the tree of life.

I have very minor comments:

I find the first part of title somewhat misleading (“Senescence across the furthest reaches of the universe?”). When I read the title, I was secretly hoping for a theoretical work discussing how senescence might occur/evolve on different planets.... Replacing the “universe” with “our planet” might be a more appropriate title

I am wondering how representative the 80 animal species analysed is of the entire animal kingdom? For example, there is only one Amphibian species on the list, and Invertebrates seem to be significantly under represented. Would adding more species from a broader sale of the animal kingdom change the authors results?

I would also be curious to know the opinion of the authors about species that change sex throughout their life (e.g. some fish species, like barramundi (*Lates calcarifer*)) where the reproductive effort/output, resource requirement/risk of predation etc.... change through the life of the organism.

Under lines 201- 203: The authors state:

“These results indicate that the patterns of senescence across animals cannot be explained by phylogenetic relatedness, assuming a brownian model of evolution. On the other hand, phylogenetic relatedness plays some role in senescence patterns across plants (Fig. S2;203 Table S3).”

I find these results very interesting, I would be curious about how the authors explain the dichotomy between plants and animals. I might have missed it, but I don't think the authors discuss this very interesting result.

Line 81:

“exclusive evolutionary theories arose both explain senescence as a by-product of” = “arose” seems to be out of place here, delete?

Line 187:

“vertbrates” change to vertebrates

Line 314:

“studying age-patterns of reproduction and mortality in tandem can we fully understanding” = change to understand

Review form: Reviewer 2

Recommendation

Accept with minor revision (please list in comments)

Scientific importance: Is the manuscript an original and important contribution to its field?

Excellent

General interest: Is the paper of sufficient general interest?

Good

Quality of the paper: Is the overall quality of the paper suitable?

Good

Is the length of the paper justified?

Yes

Should the paper be seen by a specialist statistical reviewer?

No

Do you have any concerns about statistical analyses in this paper? If so, please specify them explicitly in your report.

No

It is a condition of publication that authors make their supporting data, code and materials available - either as supplementary material or hosted in an external repository. Please rate, if applicable, the supporting data on the following criteria.

Is it accessible?

Yes

Is it clear?

Yes

Is it adequate?

Yes

Do you have any ethical concerns with this paper?

No

Comments to the Author

The paper by Roper, Capdevilla, and Salguero-Gómez presents an important overview of the current state of senescence research and the major challenges and future directions of the field.

Starting with an overview of the classical evolutionary framework for the study of senescence, the paper proceeds to present a comparative study of the pace measure of senescence for 80 animal species and 375 plant species. A majority of both plant and animal species show negligible or negative senescence. The paper then proceeds to review the mechanisms and conditions under which these species could potentially escape from senescence, before proposing to extend the classical evolutionary framework of senescence.

I enjoyed reading the paper, it was generally well-written and had exciting results and insights to provide, so I think it makes an important contribution to the field. However, I do believe that the paper can be improved in terms of presentation of some of the ideas. I also found myself looking for a stronger conclusion by the end of the paper. I agree that we should recognize that senescence is not nearly as universal as we previously believed it to be, in light of the evidence presented here and elsewhere. But what exactly do the authors propose as an extension to the evolutionary framework of senescence? The recognition that certain types of stage structure provide a disruption of early and late life tradeoffs? I agree with this too, but it doesn't quite cover the full scale of what has been parsed out in the paper. Currently the last sentence of the paper is a suggestion for what could make a holistic model to study senescence across species. Something more general, reflecting the insights from the review, might make more of an impression on your readers.

The other major point I would like to see the authors address more explicitly concerns selection gradients. I personally find it helpful that the authors show the formulas for the selection gradients as this makes it easier to discuss certain implications and fundamental insights arising from these formulas. The authors however do not show any selection gradients in the paper. I don't think they need to, such an addition might be beyond the scope of this paper. I do wonder whether showing selection gradients could provide a better link between the $m(x)$ and $l(x)$ trajectories in figure 2, rather than just showing those trajectories. On the other hand, if the authors believe the shape metric S is a sufficient proxy for a full selection gradient, I would like to see them discuss this point in a little more detail.

Title – I understand what the first half of the title is meant to convey, but I think the question mark undercuts your message a bit. It's not a question you are attempting to answer.

Abstract – Add more of a conclusion? If you drop the first sentence, something you don't really discuss in the paper, you will have more room to mention what you want people to take away from this review. Right now, it ends with why the review is relevant, not what insight it delivers.

L40 (From this): From what?

Comments below are ordered by section.

1. Classical evolutionary framework

Extensive overview, clearly written.

L110 ('force of selection'): Perhaps add a sentence on why this quantity represents the force of selection. You use this term a lot, it serves to be explicit in your definition of it.

L120 (reproductive value): Maybe define?

L135 (interacts with condition): condition of what?

L136: typo, tendency.

2. Current data – Section heading font size is not consistent

Paragraph starting at L179: You have labeled the figures as 1a, and 1b, why not refer to them as such?

L187: typo, vertebrates

Paragraph starting at L207: maybe mention that these age-specific $m(x)$ and $l(x)$ schedules were calculated from models that were originally structured by stages other than age. Also, cite figures 2a and 2b as such?

L215: This sentence is unclear to me. Independent of each other? Of the senescence result?

L219: dashes should be em-dashes

L223: This clause is not needed; you mention later that you only select matrices with larger dimensions and temporal resolutions.

3. Why and where species may not senesce

This is the section that I believe would most benefit from an improved presentation. The section starts off with the presentation of the reformulated selection gradients, so that it is clear that these depend on the stable age distribution and reproductive value. The authors propose that ecological and evolutionary mechanisms could change these distributions and that this may explain some species' escape from senescence. They then ask two questions that are somewhat vague and never explicitly answered; before they launch into a series of subsections. It took me a few readthroughs of this section before I realized what they intended; these questions are to be asked in each of the following sections as a way of investigating whether [potential mechanism influencing the $v(x)$ and $c(x)$ distributions] would ultimately provide a mechanism of escape from senescence. If that is indeed the intent, this needs to be much clearer.

The transition from these questions to subsection a) Two pillars of fitness reads as abrupt. You could add a sentence here about the types of mechanism you are about to discuss. I also think this subsection was the least clear to me in terms of what the mechanism under investigation was, if you retitled it as 'reproductive senescence' or something similar, that would prime the reader better to what you discuss.

The following two subsections had a heading that clearly set up my expectations of the mechanism to be discussed and went into an appropriate amount of detail on each of these.

L247-252: Slightly unclear, maybe split sentence?

L256: typo, than

L259 (the mechanism): what mechanism?

L264: typo, intrinsically

L274: This sentence seems incomplete, where is the subject?

L276 (these components): You haven't mentioned what these components are

L294 (age trajectories of reproduction can be uncoupled from the pattern of senescence): This came out of nowhere for me. How do you expect them to be coupled? What are the examples where reproduction is coupled / uncoupled from senescence in Figure 2?

L310: remove one instance of "may" and split this sentence

L351: "Proportional to the product"

L358 (leads to the stage structuring of populations): vague, a lot of things can potentially be stages by which populations are structured.

L358-359: (can "be" considered under the same umbrella framework): same framework as what?

L399: typo, tendencies

4. Extending the framework

As I mentioned before, I had hoped for this section to do a bit more in terms of bringing together all the insights presented in the paper and have a clearer conclusion.

L426: typo, prevalence. Also, em-dashes.

There are several typos in the reference list, please double-check it, I just scanned through it:

L476: Optimality

L480: Bacterium

L502: Williams'

L506: Antagonistic

L511: Dölling

Figures:

Appropriate and clear, although I would like to see y-axis labels in figure 2. Feel free to disregard

this idea, but could you add a separate right-hand axis for fertility? Then all the survivorship curves could start at 1 and patterns can be more easily compared.

I hope these comments are useful to you.

Review form: Reviewer 3

Recommendation

Major revision is needed (please make suggestions in comments)

Scientific importance: Is the manuscript an original and important contribution to its field?

Acceptable

General interest: Is the paper of sufficient general interest?

Good

Quality of the paper: Is the overall quality of the paper suitable?

Good

Is the length of the paper justified?

Yes

Should the paper be seen by a specialist statistical reviewer?

No

Do you have any concerns about statistical analyses in this paper? If so, please specify them explicitly in your report.

No

It is a condition of publication that authors make their supporting data, code and materials available - either as supplementary material or hosted in an external repository. Please rate, if applicable, the supporting data on the following criteria.

Is it accessible?

Yes

Is it clear?

N/A

Is it adequate?

N/A

Do you have any ethical concerns with this paper?

No

Comments to the Author

In this paper, the authors provide empirical data showing a lack of evidence for senescence across animals in plants. This has already been shown in Jones et al. (2013) "Diversity of ageing across the tree of life" published in Nature. But, here the authors highlight how population structure and reproductive values jointly determine the senescence pattern, and discuss how those quantities can be affected by stage structure, growth form and social interactions.

Overall, I am not convinced that the data reported here is of higher value than the one reported in Jones et al. (2013). Nonetheless, I think that the authors are making a very interesting (and often

overlooked) point in showing how the reproductive value can affect the pattern of senescence, based on the mathematical derivation made by Hal Caswell (1978). I believe the authors could make their point stronger by showing those two quantities in the data they got. It is easy to calculate those quantities based on population matrices, and it would clearly show how one can dissect the factors at the origin of the senescence pattern. Additionally, the authors could be clearer about the population regulation assumption made behind each prediction regarding senescence.

I apologize for any inconvenience caused by the delay in writing my review. I hope that my comments will help the authors to make their case even stronger.

MAJOR COMMENTS

1) The implication of reproductive value for senescence

lines 74-78: "The later age classes of such a cohort will therefore, all else being equal, contribute less to the ancestry of future generations (i.e. have lower reproductive value; Fisher 1930)" In this sentence, the authors refers to the effect of the stable age distribution on the reproductive value. I think this is wrong (but I am unsure given that I still have some hard time to grasp the concept of reproductive value). The reproductive value is independent of the proportion of individuals in each age-class. Actually, the reproductive value at age x can be expressed as the 'number of female offspring produced this year by mothers of age x or older' divided by the 'number of mothers that are at age x this year'. That is why there is this denominator term in the expression of the reproductive value. See the bulletin ESA, "A Simple Equation for Presenting Reproductive Value to Introductory Biology and Ecology Classes", doi:10.2307/20168257; that is the neater explanation I found on the expression of the reproductive value and indeed the number of females that survive to age x is in the denominator. Therefore even if the probability to survive to an older age is by definition smaller, this low probability of surviving to old age will actually increase the reproductive value. Given that the reproductive value is one of the main focus of this paper, I would suggest the authors to define this notion very accurately.

The implication of reproductive value for senescence (argued by Fisher 1930; and considered as such in this paragraph) is also misleading; that was actually the point made by Hamilton (1966). Charlesworth 2000 (in Genetics) sums up the situation at that time in a very neat way: "Hamilton (1966) noted that it is fallacious to use reproductive value as a measure of the effectiveness of selection as a function of age, and that a different measure should be used." "Neither of these formulae corresponds to reproductive value, and they have rather different implications for the relation between the age of effect of a gene and its impact on fitness" (compare his equation 2 vs. his equation 6)

Now I appreciate that it is possible to express those derivatives depending on the reproductive value (as in the equations 4 and 5, based on one of Caswell's derivations). The authors actually provide a very neat explanation in lines 245-252. Overall, the authors should be careful and extra-accurate when they describe the implication of reproductive value for senescence to avoid any confusion.

For instance, lines 119-120 are not crystal-clear: "equation 3 formulates how the force of selection acting against (hence the negative sign) an increase in age-specific mortality is proportional to reproductive value at age x ". This sounds wrong; for somebody who knows Hamilton's work but who is not familiar with Caswell's work (as I was), the expression of the reproductive value is not in equation 3.

The current paper changed my mind on the implication of reproductive value for senescence. I initially believed that population structure was all that matters (I only knew of Hamilton 1966 and Charlesworth 2000). This is a strength of the paper; nonetheless, the paper remains still a bit unclear because the reproductive value is not clearly defined, and its effect on senescence is

confusing in the section "Classical Evolutionary Framework of Senescence".

On the same note, I would strongly suggest the authors to show how age structure and reproductive value determine the senescence pattern (based on equation 4 and 5). It is very easy to get those quantities from the matrices they use (see doi: 10.2307/20168257 for the calculation of reproductive value). This will be very insightful to see what quantity drives the pattern of senescence, and this will considerably increase the value of the analysis shown. Additionally this will clearly illustrate how one can answer to the question "How do these distributions affect the evolutionary outcome for senescence?" (line 261) which is discussed throughout the paper.

2) Distinguishing predictions base on the assumption of exponential growth vs. density dependent growth

The authors summarize some previous results that were shown only under a specific assumption regarding the nature of the growth rate. The authors should try to be more accurate to avoid any shortcut (see the section 'Density-Dependent Population Regulation' in Moorad et al. 2019, doi:10.1016/j.tree.2019.02.006).

For instance:

lines 73-74: "There is always some non-zero probability of death, and so a cohort of individuals will decline in number over time." Actually a cohort of individuals will decline in number over time even when assuming immortality (a probability of death equal to zero) under the assumption of exponential growth; with implications for the relative strength of selection at older age vs younger age. Same in lines 90-91 that should be rephrased accordingly. Same in lines 95-96 that is true only assuming density dependence.

lines 134-136: an age-independent change in mortality will change the selection gradient if there is density dependence (because this will cause a change in age structure), but not if an exponential growth is assumed.

MINOR COMMENTS

The author use COMPADRE Plant Matrix Database and COMADRE Animal Matrix. Could the authors be clear about whether they accounted for the "persistent problems in the construction of matrix population models" highlighted by Kendall et al. 2019 in Ecological Modelling (doi:10.1016/j.ecolmodel.2019.03.011).

The authors could consider changing the title of the paper and removing "Senescence across the furthest reaches of the universe?". It means nothing to me; I guess it will confuse other readers.

lines 36-38: "Most ageing research likely stems from human desire to increase human lifespan and health span. This desire requires understanding" This does not seem very informative and relevant for the rest of the paper. Perhaps the authors could consider dropping that sentence.

lines 39-40: "understanding why different factors, rather than age per se, alter the sensitivity of fitness to age-related changes in mortality and reproduction". This sentence implies that previous research has focused on "understanding why age alters the sensitivity of fitness to age-related changes in mortality and reproduction", which does not sound right at all. Additionally, as argued in this manuscript, there have been extensive research on how extrinsic mortality alters the sensitivity of fitness to age-related changes in mortality. So it is not novel at all to investigate those factors that alter the strength of selection on such senescence-related mutations.

lines 64-65: "The central logic common to these theories argues that the force of natural selection weakens with age". This is not entirely true given that some theories argue that antagonistic pleiotropy can favour certain genes (as written in lines 66-67: "or [selection] favours these genes"),

and antagonistic pleiotropy could lead to senescence without the need for weak selection at older age; e.g., if the benefit at earlier age is massive.

line 141: "senescence has not evolved adaptively"; this is not entirely true. The authors indeed argue that senescence can evolve via antagonistic pleiotropy. In that case, this is adaptive evolution.

lines 190 and 211: "positive senescence" is not defined. I would suggest the authors to employ the term "actuarial senescence" instead, for consistently. Alternatively, the authors could employ positive senescence throughout the manuscript.

line 218: should it be "long-term individually-based"?

lines 225-226: "We countered these limitations by using a thorough selection criteria". What do the authors get without using this criterion? Do they get qualitatively similar results?

Decision letter (RSPB-2021-0851.R0)

08-Jun-2021

Dear Dr Roper:

Your manuscript has now been peer reviewed and the reviewers' comments (not including confidential comments to the Editor) are included at the end of this email for your reference. As you will see, the reviewers have raised some concerns with your manuscript and, because with some serious revision all think that the paper would be a very positive contribution to an important literature, I would like to invite you to revise your manuscript to address them. If I could add my own, pretty trivial, comment, I agree with the reviewers that the title needs changing - the first part adds nothing and is actually misleading. How about the simpler "Senescence: why and where selection gradients might not decline with age"?

We do not allow multiple rounds of revision so we urge you to make every effort to fully address all of the comments at this stage. If deemed necessary your manuscript will be sent back to one or more of the original reviewers for assessment. If the original reviewers are not available we may invite new reviewers. Please note that we cannot guarantee eventual acceptance of your manuscript at this stage.

Research ethics:

Use of animals and field studies:

It is a condition of publication that you make available the data and research materials supporting the results in the article (<https://royalsociety.org/journals/authors/author-guidelines/#data>). Datasets should be deposited in an appropriate publicly available repository and details of the associated accession number, link or DOI to the datasets must be included in the Data Accessibility section of the article (<https://royalsociety.org/journals/ethics-policies/data-sharing-mining/>). Reference(s) to datasets should also be included in the reference list of the article with DOIs (where available).

Please submit a copy of your revised paper within three weeks. If we do not hear from you within this time your manuscript will be rejected. If you are unable to meet this deadline please let us know as soon as possible, as we may be able to grant a short extension.

Best wishes,
 mailto: proceedingsb@royalsociety.org

Reviewer(s)' Comments to Author:

Referee: 1

Comments to the Author(s)

The manuscript by Roper et al. titled:

"Senescence across the furthest reaches of the universe? Why and where selection gradients might not decline with age." is beautifully written, an absolute pleasure to read.

The authors first provide an excellent review of the senescence literature and provide an overview of the empirical and theoretical mechanisms that have been proposed to drive the variability of senescence patterns. Then they offer a framework to support their theory, that only by deciphering the underlying factors of age-related changes in mortality and reproduction can one understand why aging has evolved across the tree of life.

I have very minor comments:

I find the first part of title somewhat misleading ("Senescence across the furthest reaches of the universe?). When I read the title, I was secretly hoping for a theoretical work discussing how senescence might occur/evolve on different planets.... Replacing the "universe" with "our planet" might be a more appropriate title

I am wondering how representative the 80 animal species analysed is of the entire animal kingdom? For example, there is only one Amphibian species on the list, and Invertebrates seem to be significantly under represented. Would adding more species from a broader sale of the animal kingdom change the authors results?

I would also be curious to know the opinion of the authors about species that change sex throughout their life (e.g. some fish species, like barramundi (*Lates calcarifer*)) where the reproductive effort/output, resource requirement/risk of predation etc.... change through the life of the organism.

Under lines 201- 203: The authors state:

"These results indicate that the patterns of senescence across animals cannot be explained by phylogenetic relatedness, assuming a brownian model of evolution. On the other hand, phylogenetic relatedness plays some role in senescence patterns across plants (Fig. S2;203 Table S3)."

I find these results very interesting, I would be curious about how the authors explain the dichotomy between plants and animals. I might have missed it, but I don't think the authors discuss this very interesting result.

Line 81:

"exclusive evolutionary theories arose both explain senescence as a by-product of" = "arose" seems to be out of place here, delete?

Line 187:

"vertbrates" change to vertebrates

Line 314:

"studying age-patterns of reproduction and mortality in tandem can we fully understanding" = change to understand

Referee: 2

Comments to the Author(s)

The paper by Roper, Capdevilla, and Salguero-Gómez presents an important overview of the current state of senescence research and the major challenges and future directions of the field. Starting with an overview of the classical evolutionary framework for the study of senescence, the paper proceeds to present a comparative study of the pace measure of senescence for 80 animal species and 375 plant species. A majority of both plant and animal species show negligible or negative senescence. The paper then proceeds to review the mechanisms and conditions under which these species could potentially escape from senescence, before proposing to extend the classical evolutionary framework of senescence.

I enjoyed reading the paper, it was generally well-written and had exciting results and insights to provide, so I think it makes an important contribution to the field. However, I do believe that the paper can be improved in terms of presentation of some of the ideas. I also found myself looking for a stronger conclusion by the end of the paper. I agree that we should recognize that senescence is not nearly as universal as we previously believed it to be, in light of the evidence presented here and elsewhere. But what exactly do the authors propose as an extension to the evolutionary framework of senescence? The recognition that certain types of stage structure provide a disruption of early and late life tradeoffs? I agree with this too, but it doesn't quite cover the full scale of what has been parsed out in the paper. Currently the last sentence of the paper is a suggestion for what could make a holistic model to study senescence across species. Something more general, reflecting the insights from the review, might make more of an impression on your readers.

The other major point I would like to see the authors address more explicitly concerns selection gradients. I personally find it helpful that the authors show the formulas for the selection gradients as this makes it easier to discuss certain implications and fundamental insights arising from these formulas. The authors however do not show any selection gradients in the paper. I don't think they need to, such an addition might be beyond the scope of this paper. I do wonder whether showing selection gradients could provide a better link between the $m(x)$ and $l(x)$ trajectories in figure 2, rather than just showing those trajectories. On the other hand, if the authors believe the shape metric S is a sufficient proxy for a full selection gradient, I would like to see them discuss this point in a little more detail.

Title - I understand what the first half of the title is meant to convey, but I think the question mark undercuts your message a bit. It's not a question you are attempting to answer.

Abstract - Add more of a conclusion? If you drop the first sentence, something you don't really discuss in the paper, you will have more room to mention what you want people to take away from this review. Right now, it ends with why the review is relevant, not what insight it delivers.
L40 (From this): From what?

Comments below are ordered by section.

1. Classical evolutionary framework

Extensive overview, clearly written.

L110 ('force of selection'): Perhaps add a sentence on why this quantity represents the force of selection. You use this term a lot, it serves to be explicit in your definition of it.

L120 (reproductive value): Maybe define?

L135 (interacts with condition): condition of what?

L136: typo, tendency.

2. Current data - Section heading font size is not consistent

Paragraph starting at L179: You have labeled the figures as 1a, and 1b, why not refer to them as such?

L187: typo, vertebrates

Paragraph starting at L207: maybe mention that these age-specific $m(x)$ and $l(x)$ schedules were calculated from models that were originally structured by stages other than age. Also, cite figures 2a and 2b as such?

L215: This sentence is unclear to me. Independent of each other? Of the senescence result?

L219: dashes should be em-dashes

L223: This clause is not needed; you mention later that you only select matrices with larger dimensions and temporal resolutions.

3. Why and where species may not senesce

This is the section that I believe would most benefit from an improved presentation. The section starts off with the presentation of the reformulated selection gradients, so that it is clear that these depend on the stable age distribution and reproductive value. The authors propose that ecological and evolutionary mechanisms could change these distributions and that this may explain some species' escape from senescence. They then ask two questions that are somewhat vague and never explicitly answered; before they launch into a series of subsections. It took me a few readthroughs of this section before I realized what they intended; these questions are to be asked in each of the following sections as a way of investigating whether [potential mechanism influencing the $v(x)$ and $c(x)$ distributions] would ultimately provide a mechanism of escape from senescence. If that is indeed the intent, this needs to be much clearer.

The transition from these questions to subsection a) Two pillars of fitness reads as abrupt. You could add a sentence here about the types of mechanism you are about to discuss. I also think this subsection was the least clear to me in terms of what the mechanism under investigation was, if you retitled it as 'reproductive senescence' or something similar, that would prime the reader better to what you discuss.

The following two subsections had a heading that clearly set up my expectations of the mechanism to be discussed and went into an appropriate amount of detail on each of these.

L247-252: Slightly unclear, maybe split sentence?

L256: typo, than

L259 (the mechanism): what mechanism?

L264: typo, intrinsically

L274: This sentence seems incomplete, where is the subject?

L276 (these components): You haven't mentioned what these components are

L294 (age trajectories of reproduction can be uncoupled from the pattern of senescence): This came out of nowhere for me. How do you expect them to be coupled? What are the examples where reproduction is coupled / uncoupled from senescence in Figure 2?

L310: remove one instance of "may" and split this sentence

L351: "Proportional to the product"

L358 (leads to the stage structuring of populations): vague, a lot of things can potentially be stages by which populations are structured.

L358-359: (can "be" considered under the same umbrella framework): same framework as what?

L399: typo, tendencies

4. Extending the framework

As I mentioned before, I had hoped for this section to do a bit more in terms of bringing together all the insights presented in the paper and have a clearer conclusion.

L426: typo, prevalence. Also, em-dashes.

There are several typos in the reference list, please double-check it, I just scanned through it:

L476: Optimality

L480: Bacterium

L502: Williams'

L506: Antagonistic

L511: Dölling

Figures:

Appropriate and clear, although I would like to see y-axis labels in figure 2. Feel free to disregard this idea, but could you add a separate right-hand axis for fertility? Then all the survivorship curves could start at 1 and patterns can be more easily compared.

I hope these comments are useful to you.

Referee: 3

Comments to the Author(s)

In this paper, the authors provide empirical data showing a lack of evidence for senescence across animals in plants. This has already been shown in Jones et al. (2013) "Diversity of ageing across the tree of life" published in Nature. But, here the authors highlight how population structure and reproductive values jointly determine the senescence pattern, and discuss how those quantities can be affected by stage structure, growth form and social interactions.

Overall, I am not convinced that the data reported here is of higher value than the one reported in Jones et al. (2013). Nonetheless, I think that the authors are making a very interesting (and often overlooked) point in showing how the reproductive value can affect the pattern of senescence, based on the mathematical derivation made by Hal Caswell (1978). I believe the authors could make their point stronger by showing those two quantities in the data they got. It is easy to calculate those quantities based on population matrices, and it would clearly show how one can dissect the factors at the origin of the senescence pattern. Additionally, the authors could be clearer about the population regulation assumption made behind each prediction regarding senescence.

I apologize for any inconvenience caused by the delay in writing my review. I hope that my comments will help the authors to make their case even stronger.

MAJOR COMMENTS

1) The implication of reproductive value for senescence

lines 74-78: "The later age classes of such a cohort will therefore, all else being equal, contribute less to the ancestry of future generations (i.e. have lower reproductive value; Fisher 1930)" In this sentence, the authors refers to the effect of the stable age distribution on the reproductive value. I think this is wrong (but I am unsure given that I still have some hard time to grasp the concept of reproductive value). The reproductive value is independent of the proportion of individuals in each age-class. Actually, the reproductive value at age x can be expressed as the 'number of female offspring produced this year by mothers of age x or older' divided by the 'number of mothers that are at age x this year'. That is why there is this denominator term in the expression of the reproductive value. See the bulletin ESA, "A Simple Equation for Presenting Reproductive Value to Introductory Biology and Ecology Classes", doi:10.2307/20168257; that is the neater explanation I found on the expression of the reproductive value and indeed the number of females that survive to age x is in the denominator. Therefore even if the probability to survive to an older age is by definition smaller, this low probability of surviving to old age will actually increase the reproductive value. Given that the reproductive value is one of the main focus of this paper, I would suggest the authors to define this notion very accurately.

The implication of reproductive value for senescence (argued by Fisher 1930; and considered as such in this paragraph) is also misleading; that was actually the point made by Hamilton (1966). Charlesworth 2000 (in Genetics) sums up the situation at that time in a very neat way: "Hamilton (1966) noted that it is fallacious to use reproductive value as a measure of the effectiveness of selection as a function of age, and that a different measure should be used." "Neither of these formulae corresponds to reproductive value, and they have rather different implications for the

relation between the age of effect of a gene and its impact on fitness" (compare his equation 2 vs. his equation 6)

Now I appreciate that it is possible to express those derivatives depending on the reproductive value (as in the equations 4 and 5, based on one of Caswell's derivations). The authors actually provide a very neat explanation in lines 245-252. Overall, the authors should be careful and extra-accurate when they describe the implication of reproductive value for senescence to avoid any confusion.

For instance, lines 119-120 are not crystal-clear: "equation 3 formulates how the force of selection acting against (hence the negative sign) an increase in age-specific mortality is proportional to reproductive value at age x ". This sounds wrong; for somebody who knows Hamilton's work but who is not familiar with Caswell's work (as I was), the expression of the reproductive value is not in equation 3.

The current paper changed my mind on the implication of reproductive value for senescence. I initially believed that population structure was all that matters (I only knew of Hamilton 1966 and Charlesworth 2000). This is a strength of the paper; nonetheless, the paper remains still a bit unclear because the reproductive value is not clearly defined, and its effect on senescence is confusing in the section "Classical Evolutionary Framework of Senescence".

On the same note, I would strongly suggest the authors to show how age structure and reproductive value determine the senescence pattern (based on equation 4 and 5). It is very easy to get those quantities from the matrices they use (see doi: 10.2307/20168257 for the calculation of reproductive value). This will be very insightful to see what quantity drives the pattern of senescence, and this will considerably increase the value of the analysis shown. Additionally this will clearly illustrate how one can answer to the question "How do these distributions affect the evolutionary outcome for senescence?" (line 261) which is discussed throughout the paper.

2) Distinguishing predictions base on the assumption of exponential growth vs. density dependent growth

The authors summarize some previous results that were shown only under a specific assumption regarding the nature of the growth rate. The authors should try to be more accurate to avoid any shortcut (see the section 'Density-Dependent Population Regulation' in Moorad et al. 2019, doi:10.1016/j.tree.2019.02.006).

For instance:

lines 73-74: "There is always some non-zero probability of death, and so a cohort of individuals will decline in number over time." Actually a cohort of individuals will decline in number over time even when assuming immortality (a probability of death equal to zero) under the assumption of exponential growth; with implications for the relative strength of selection at older age vs younger age. Same in lines 90-91 that should be rephrased accordingly. Same in lines 95-96 that is true only assuming density dependence.

lines 134-136: an age-independent change in mortality will change the selection gradient if there is density dependence (because this will cause a change in age structure), but not if an exponential growth is assumed.

MINOR COMMENTS

The author use COMPADRE Plant Matrix Database and COMADRE Animal Matrix. Could the authors be clear about whether they accounted for the "persistent problems in the construction of matrix population models" highlighted by Kendall et al. 2019 in Ecological Modelling (doi:10.1016/j.ecolmodel.2019.03.011).

The authors could consider changing the title of the paper and removing "Senescence across the furthest reaches of the universe?". It means nothing to me; I guess it will confuse other readers.

lines 36-38: "Most ageing research likely stems from human desire to increase human lifespan and health span. This desire requires understanding" This does not seem very informative and relevant for the rest of the paper. Perhaps the authors could consider dropping that sentence.

lines 39-40: "understanding why different factors, rather than age per se, alter the sensitivity of fitness to age-related changes in mortality and reproduction". This sentence implies that previous research has focused on "understanding why age alters the sensitivity of fitness to age-related changes in mortality and reproduction", which does not sound right at all. Additionally, as argued in this manuscript, there have been extensive research on how extrinsic mortality alters the sensitivity of fitness to age-related changes in mortality. So it is not novel at all to investigate those factors that alter the strength of selection on such senescence-related mutations.

lines 64-65: "The central logic common to these theories argues that the force of natural selection weakens with age". This is not entirely true given that some theories argue that antagonistic pleiotropy can favour certain genes (as written in lines 66-67: "or [selection] favours these genes"), and antagonistic pleiotropy could lead to senescence without the need for weak selection at older age; e.g., if the benefit at earlier age is massive.

line 141: "senescence has not evolved adaptively"; this is not entirely true. The authors indeed argue that senescence can evolve via antagonistic pleiotropy. In that case, this is adaptive evolution.

lines 190 and 211: "positive senescence" is not defined. I would suggest the authors to employ the term "actuarial senescence" instead, for consistently. Alternatively, the authors could employ positive senescence throughout the manuscript.

line 218: should it be "long-term individually-based"?

lines 225-226: "We countered these limitations by using a thorough selection criteria". What do the authors get without using this criterion? Do they get qualitatively similar results?

Author's Response to Decision Letter for (RSPB-2021-0851.R0)

See Appendix A.

Decision letter (RSPB-2021-0851.R1)

30-Jun-2021

Dear Mr Roper

I am pleased to inform you that your manuscript entitled "Senescence: why and where selection gradients might not decline with age." has been accepted for publication in Proceedings B.

You can expect to receive a proof of your article from our Production office in due course, please check your spam filter if you do not receive it. PLEASE NOTE: you will be given the exact page

length of your paper which may be different from the estimation from Editorial and you may be asked to reduce your paper if it goes over the 10 page limit.

If you are likely to be away from e-mail contact during this period, let us know. Due to rapid publication and an extremely tight schedule, if comments are not received, we may publish the paper as it stands.

Data Accessibility section

Open access

You are invited to opt for open access via our author pays publishing model. Payment of open access fees will enable your article to be made freely available via the Royal Society website as soon as it is ready for publication. For more information about open access publishing please visit our website at http://royalsocietypublishing.org/site/authors/open_access.xhtml.

The open access fee is £1,700 per article (plus VAT for authors within the EU). If you wish to opt for open access then please let us know as soon as possible.

Paper charges

Sincerely,
Proceedings B
<mailto:proceedingsb@royalsociety.org>

Appendix A

Dear Proceedings B Editorial Team,

Thank you for the opportunity to revise our manuscript. We have now addressed all of the editor and reviewer's concerns which we detail below. The main changes to our manuscript we present are:

- A re-structured section 3 & 4 to clarify the direction of section 3 (“Why and where species may not senesce”) when addressing the different factors that might alter predicted patterns of senescence, and to add a stronger conclusion in section 4 (“Extending the classical evolutionary framework of senescence”) (R2).
- A re-written abstract to address the different concerns of R2 & R3.
- A secondary y-axis displaying age-specific reproduction ($m(x)$) in Figure 2, now Figures 2a and 2c, which allows clearer visualisation of both survivorship and reproductive trajectories on the same graph, as suggested by R2.
- Stable (st)age and reproductive value distributions for species in Figures 1 & 2 (original manuscript) and the supplementary information to demonstrate the importance of these two metrics for the evolution of senescence, and to our argument within this manuscript (R2 & R3).

Given the re-structuring of sections 3 & 4 and the addition of 2 new figures, we have also streamlined some sections to meet the editorial policies of Proceedings B. To facilitate the assessment of our revised work we have kept a tracked changes version of the manuscript below our responses to the reviewers in this file. Our responses below are preceded by “R#>”, where # states the numeric in chronological order, and in blue font. References to line numbers to demonstrate our responses represent the line numbers from the ‘FinalVersion.doc’ document that is our new main manuscript file.

We would like to thank the editor for the suggested simpler title, which we have now used.

We hope that the revisions we have made will prove satisfactory for publication in Proceedings B.

Yours sincerely,

Mark Roper, on behalf of all co-authors
Corresponding Author RSPB-2021-0851

Response to Reviewers

Referee 1

The manuscript by Roper et al. titled: “Senescence across the furthest reaches of the universe? Why and where selection gradients might not decline with age.” is beautifully written, an absolute pleasure to read. The authors first provide an excellent review of the senescence literature and provide an overview of the empirical and theoretical mechanisms that have been proposed to drive the variability of senescence patterns. Then they offer a framework to support their theory, that only by deciphering the underlying factors of age-related changes in mortality and reproduction can one understand why aging has evolved across the tree of life.

R1> We are grateful to the reviewer for their kind comments and for taking the time to review our manuscript.

I have very minor comments: I find the first part of title somewhat misleading (“Senescence across the furthest reaches of the universe?”). When I read the title, I was secretly hoping for a theoretical work discussing how senescence might occur/evolve on different planets.... Replacing the “universe” with “our planet” might be a more appropriate title

R2> Changing the manuscript title was something noted by all reviewers and the editor, and so we have adjusted accordingly to “*Senescence: why and where selection gradients might not decline with age*”.

I am wondering how representative the 80-animal species analysed is of the entire animal kingdom? For example, there is only one Amphibian species on the list, and Invertebrates seem to be significantly under represented. Would adding more species from a broader sale of the animal kingdom change the authors results?

R3> We agree with the reviewer. Our study is not as representative as we would like – but is the most comprehensive yet. To emphasise this view, which we fully share with the reviewer, we already talked about data representability and biases in lines 444-445 of the original submitted manuscript with reference 103. We have added onto this to emphasise the importance of this comment from the reviewer:

L416-419: High-resolution demographic data on some major groups of species are still yet limited¹⁰³. We hope that our work contributes to stimulating the next generation of demographers to explore these under-represented corners of biodiversity.

Unfortunately, we (i.e. the discipline as a whole, not just us) currently do not have open access to data from these under-represented taxa. We hope that our work can contribute to inspiring the next generation of demographers to go out and explore these species. Given our main result is that patterns of senescence are diverse and require us to consider alternative life history and ecological parameters from those previously considered, we believe such further exploration would only reaffirm our thesis.

I would also be curious to know the opinion of the authors about species that change sex throughout their life (e.g. some fish species, like barramundi (*Lates calcarifer*)) where the reproductive effort/output, resource requirement/risk of predation etc.... change through the life of the organism.

R4> We thank the reviewer for this thought-provoking idea. Our immediate thoughts are speculative and so we have avoided including them in the manuscript. As we try to make clear in our manuscript, all mechanisms affecting senescence must influence the stable age/reproductive value distributions. Sex ratio is key for reproductive value (RV), and there could be potential for some very interesting interactions between age and sex with respect to RV, as the work by Biezychudek with Jack-in-the-pulpit (1982). The reviewer’s idea has also got us thinking about interesting consequences for plastic senescence phenotypes and sexually antagonistic senescence alleles, which fall outside of the scope of this much more focused piece. Thank you!

Biezychudek, P. 1982. The Demography of Jack-in-the-Pulpit, a Forest Perennial that Changes Sex. *Ecological Monographs*. **52**, 335-351.

Under lines 201- 203: The authors state: “These results indicate that the patterns of senescence across animals cannot be explained by phylogenetic relatedness, assuming a brownian model of evolution. On the other hand, phylogenetic relatedness plays some role in senescence patterns across plants (Fig. S2;203 Table S3).”

I find these results very interesting, I would be curious about how the authors explain the dichotomy between plants and animals. I might have missed it, but I don’t think the authors discuss this very interesting result.

R6> Thank you for the opportunity to discuss this in further detail, even if only in a speculative manner. We believe that the reason this is so is that gymnosperms and angiosperms are known to have rather different demographies, with the former being more constrained due to their mostly monopodial growth forms, while angiosperms are much more diverse in their bauplans... and thus their demographies. Although the set of animal species we analyse here represents the most comprehensive set of demographies analysed to date -that we are aware of- from an ageing perspective, it is possible that their bauplans are more variable and not necessarily limited by such strong taxonomic bounds as it seems to be the case in plants. However, we find this argument highly speculative, and as such, we prefer not to add it to this piece (we have added our hesitation in L170-172). We do think that there’s more in these results, and would like to explore these further as a separate manuscript in the future. Some key references that support our explanation for plants include:

Baudisch, A. *et al.* 2013. The pace and shape of senescence in angiosperms. *J. Ecol.* **101**, 596-606.

Bernard, C. *et al.* 2020. Testing Finch's hypothesis: The role of organismal modularity on the escape from actuarial senescence. *F. Ecol.* **34**, 88-106.

L170-172: *More thorough phylogenetic analyses are required, however, to rule out the effects of ancestral constraint on the patterns of senescence in animals.*

Line 81:

“exclusive evolutionary theories arose both explain senescence as a by-product of” = “arose” seems to be out of place here, delete?

Line 187:

“vertebrates” change to vertebrates

Line 314:

“studying age-patterns of reproduction and mortality in tandem can we fully understanding” = change to understand

R7> Many thanks for spotting these three typos, we have corrected accordingly.

Referee 2

The paper by Roper, Capdevila, and Salguero-Gómez presents an important overview of the current state of senescence research and the major challenges and future directions of the field. Starting with an overview of the classical evolutionary framework for the study of senescence, the paper proceeds to present a comparative study of the pace measure of senescence for 80 animal species and 375 plant species. A majority of both plant and animal species show negligible or negative senescence. The paper then proceeds to review the mechanisms and conditions under which these species could potentially escape from senescence, before proposing to extend the classical evolutionary framework of senescence.

I enjoyed reading the paper, it was generally well-written and had exciting results and insights to provide, so I think it makes an important contribution to the field. However, I do believe that the paper can be improved in terms of presentation of some of the ideas. I also found myself looking for a stronger conclusion by the end of the paper. I agree that we should recognize that senescence is not nearly as universal as we previously believed it to be, in light of the evidence presented here and elsewhere. But what exactly do the authors propose as an extension to the evolutionary framework of senescence? The recognition that certain types of stage structure provide a disruption of early and late life tradeoffs? I agree with this too, but it doesn't quite cover the full scale of what has been parsed out in the paper. Currently the last sentence of the paper is a suggestion for what could make a holistic model to study senescence across species. Something more general, reflecting the insights from the review, might make more of an impression on your readers.

R8> We thank the reviewer for their comments and taking the time to review our manuscript. To address their concerns, we have re-written section 4 (“Extending the classical evolutionary framework of senescence”), to provide a stronger conclusion that reflects the general insights we hoped to convey.

The other major point I would like to see the authors address more explicitly concerns selection gradients. I personally find it helpful that the authors show the formulas for the selection gradients as this makes it easier to discuss certain implications and fundamental insights arising from these formulas. The authors however do not show any selection gradients in the paper. I don't think they need to, such an addition might be beyond the scope of this paper. I do wonder whether showing selection gradients could provide a better link between the $m(x)$ and $l(x)$ trajectories in figure 2, rather than just showing those trajectories. On the other hand, if the authors believe the shape metric S is a sufficient proxy for a full selection gradient, I would like to see them discuss this point in a little more detail.

R9 > We agree with the reviewer that this would be helpful, and, along with the suggestions from reviewer 3, we have now added the two main components of the selection gradient (reproductive value and the stable age distribution) to create two new figures, which are discussed in detail in section 3. We have also added the same format for all of our examined species in the supplementary information. We have embedded our interpretation of our findings into Section 3 of the manuscript, specifically in L227-235. We do not believe the shape metric S is suitable for this purpose, and agree with both reviewers that adding this extra information will improve the manuscript considerably.

L227-235: *Figures 2b and 2d exemplify the importance of reproductive value and population structure for explaining divergent patterns of senescence (see also Fig. S3). All species (except Pinus sylvestris; Fig. 2d) display reproductive value and stable age distributions that match their classification of actuarial senescence. Species that display positive actuarial senescence have distributions biased to earlier (st)ages, and those that do not have distributions that are more heavily skewed towards later (st)ages. Our assumption of each population sampled being at a stable state at the time of data collection provides the most intuitive reason why the patterns (including for Pinus sylvestris) may not match. It is clear from our analysis that reproductive value and population structure are key to understanding variation in patterns of senescence.*

Title – I understand what the first half of the title is meant to convey, but I think the question mark undercuts your message a bit. It's not a question you are attempting to answer.

R10> Changing the manuscript title was something noted by all reviewers and the editor, and so we have adjusted accordingly to “*Senescence: why and where selection gradients might not decline with age*”.

Abstract – Add more of a conclusion? If you drop the first sentence, something you don't really discuss in the paper, you will have more room to mention what you want people to take away from this review. Right now, it ends with why the review is relevant, not what insight it delivers.

R11> We have removed this sentence, and, along with guidance also from the reviewer 3, have revised the abstract to meet all the concerns of both reviewers. Our abstract now focuses more on the insights we want people to take away from our review.

L40 (From this): From what?

R11> Thank you for spotting this, we have added the subject ‘From this understanding,’ into our modified abstract.

Comments below are ordered by section.

1. Classical evolutionary framework

Extensive overview, clearly written.

L110 (‘force of selection’): Perhaps add a sentence on why this quantity represents the force of selection. You use this term a lot, it serves to be explicit in your definition of it.

R12> Excellent suggestion. We have now clarified why this term represents a force of selection in L93-95

L93-95: *The resulting quantities represent ‘forces of selection’ on age-specific vital rates³. The larger the absolute value of the quantity, the stronger the response of selection to a given change.* ^{7,12}

L120 (reproductive value): Maybe define?

R13> This sentence was re-written after comments from reviewer 3 and no longer contains reproductive value, which we define later when first mentioned (L214-215)

L214-215: *$v(x)$ is reproductive value⁵ (the average contribution to the ancestry of future generations)*

L135 (interacts with condition): condition of what?

R14> We have changed this to ‘physiological condition’ to be more explicit (L111-112)

L136: typo, tendency.

R15> Thank you for spotting this, the typo has been corrected.

2. Current data – Section heading font size is not consistent

R16> Thank you for spotting this, we have change to make consistent.

Paragraph starting at L179: You have labeled the figures as 1a, and 1b, why not refer to them as such?

R17> We have changed our labelling to be more accurate throughout the manuscript.

L187: typo, vertebrates

R18> Many thanks, corrected.

Paragraph starting at L207: maybe mention that these age-specific $m(x)$ and $l(x)$ schedules were calculated from models that were originally structured by stages other than age. Also, cite figures 2a and 2b as such?

R19> We agree with the reviewer that this is an important point to make, and included it earlier on in this section to make the methodology clearer from the start (L144-146):

L144-156: Finally, we use derived life tables³⁹ from both age and stage-based models (See Methods) to quantify age-specific reproduction rates ($m(x)$) to evaluate whether they match their patterns of actuarial senescence.

We have also changed our labelling to be more accurate throughout the manuscript.

L215: This sentence is unclear to me. Independent of each other? Of the senescence result?

R20 > The point we were hoping to make is that survivorship and reproductive trajectories appear to be able to follow divergent trajectories *i.e.* many species appear to display senescence for one component but not the other. We hope to have clarified this in L186-187.

L186-187: *It appears from our study species that survivorship and reproduction can follow independent age-based trajectories.*

L219: dashes should be em-dashes

R21> Many thanks, corrected.

L223: This clause is not needed; you mention later that you only select matrices with larger dimensions and temporal resolutions.

R22> We thank the reviewer for spotting this and have removed the clause.

3. Why and where species may not senesce

This is the section that I believe would most benefit from an improved presentation. The section starts off with the presentation of the reformulated selection gradients, so that it is clear that these depend on the stable age distribution and reproductive value. The authors propose that ecological and evolutionary mechanisms could change these distributions and that this may explain some species' escape from senescence. They then ask two questions that are somewhat vague and never explicitly answered; before they launch into a series of subsections. It took me a few readthroughs of this section before I realized what they intended; these questions are to be asked in each of the following sections as a way of investigating whether [potential mechanism influencing the $v(x)$ and $c(x)$ distributions] would ultimately provide a mechanism of escape from senescence. If that is indeed the intent, this needs to be much clearer.

R23> We agree with the reviewer that the previous version was not as clear as we intended. We are glad that the reviewer was able to realise our intentions, but agree that the direction of section 3 needed to be made much clearer. We hope to have rectified this issue in a section we have added below the two questions (L243-247).

L243-247: *Here, we consider how (i) the many components of reproductive senescence and trade-offs between survival and reproduction, (ii) the stage structuring of populations, and finally (iii) the spatial structuring of*

populations can act as ecological and evolutionary mechanisms that alter reproductive value and stable age distributions in ways that drive the evolution of senescence in different directions than those predicted from the classical evolutionary framework.

The transition from these questions to subsection a) Two pillars of fitness reads as abrupt. You could add a sentence here about the types of mechanism you are about to discuss. I also think this subsection was the least clear to me in terms of what the mechanism under investigation was, if you retitled it as ‘reproductive senescence’ or something similar, that would prime the reader better to what you discuss.

R24> We hope to have resolved this issue with the section noted in R23. We have also changed the title of section 3a to Reproductive senescence – we agree with the reviewer that this primes the reader better.

The following two subsections had a heading that clearly set up my expectations of the mechanism to be discussed and went into an appropriate amount of detail on each of these.

R25> Many thanks. We’re glad we were able to do a better job with these two sections.

L247-252: Slightly unclear, maybe split sentence?

R26> We have split the sentence to make it clearer.

L256: typo, than

R27> Thank you for spotting, corrected.

L259 (the mechanism): what mechanism?

R28> We have modified this sentence to clarify this with reference to the previous sentence by changing ‘the mechanism’ to ‘the ecological and/or evolutionary mechanism’ (L239-241).

L239-241: *How does the ecological and/or evolutionary mechanism alter the reproductive value and stable age distributions of the population from that expected of a standard age structured population?*

L264: typo, intrinsically

R29> Thank you for spotting, corrected.

L274: This sentence seems incomplete, where is the subject?

R30> We have edited the sentence to clarify (L261-262)

L274-275: *As reviewed by Lemaître & Gaillard⁵¹, quantifying reproductive senescence is complicated by the multitude of different components that can comprise the reproductive schedule.*

L276 (these components): You haven’t mentioned what these components are

R31> We have added a clause (L262-264) to demonstrate some of the many different components of the reproductive schedule.

L262-264: *When broken down into these components, from egg production to juvenile survival and everything in between, heterogeneous rates of reproductive senescence can be revealed.*

L294 (age trajectories of reproduction can be uncoupled from the pattern of senescence): This came out of nowhere for me. How do you expect them to be coupled? What are the examples where reproduction is coupled / uncoupled from senescence in Figure 2?

R32> We expected in general that species that displayed actuarial senescence should also display reproductive decline with age. This expectation is based of Williams’ (1957) prediction that there should be synchrony in all

components of senescence. We discuss in L207-216 of the originally submitted manuscript (now L178-187) how the species in Figure 2 do not seem to display this synchrony. Divergent trajectories are also evident in the supplementary information figures.

L178-187: *Patterns of reproduction ($m(x)$) are diverse and appear somewhat independent of whether the examined species displays actuarial senescence (Fig.2a; Fig.2c; Fig.S3). In plants, for example, both the scots pine and the great laurel display actuarial senescence (Fig. 1b), but their reproductive outputs do not decline with age (Fig. 2c). This pattern is contrasting to both examples of animals displaying positive senescence, where the moose (*Alces alces*) and water flea (*Daphnia pulex*) also display reproductive decline with age (Fig. 2a). The flatweed exhibits negligible actuarial senescence and a relatively constant $m(x)$ trajectory, whereas the long-wristed hermit crab and the sugar pine also display negligible actuarial senescence but have increasing $m(x)$ trajectories. It appears from our study species that survivorship and reproduction can follow independent age-based trajectories.*

L310: remove one instance of “may” and split this sentence

R33> We agree this sentence was poorly written and have re-written to clarify, now in L293-296:

L293-296: *Links can then be made to how reproductive and survivorship trajectories should co-evolve when constrained by physiological trade-offs^{65,66}, unveiling mechanisms behind the variation in synchrony of actuarial and reproductive senescence across species.*

L351: “Proportional to the product”

R34> Thank you for spotting, this has been corrected.

L358 (leads to the stage structuring of populations): vague, a lot of things can potentially be stages by which populations are structured.

L358-359: (can “be” considered under the same umbrella framework): same framework as what?

R35> (both comments above) We agree with the reviewer that this sentence was unclear, and have decided to remove it from the manuscript.

L399: typo, tendencies

R36> Thank you for spotting, this has been corrected.

4. Extending the framework

As I mentioned before, I had hoped for this section to do a bit more in terms of bringing together all the insights presented in the paper and have a clearer conclusion.

R37> We agree with reviewer that our conclusion could (and should) have been stronger, and hope to have done so in our revised version of section 4. Specifically, we now highlight the main insight we wanted readers to take away from our manuscript; that senescence can be understood through selection gradients, and selection gradients can be understood through reproductive value and stable age distributions. These distributions can take different shapes than those predicted from classical evolutionary theory due to different ecological and evolutionary mechanisms not considered by those works, and this explains why species across the tree of life show such vastly different patterns of ageing. We articulate this argument in our new concluding paragraph L425-439.

L425-439: *We show in our review of theory and analyses (Fig. 2b; Fig.2d; Fig. S3) that reproductive value and population structure are key to understanding senescence. The two-question process outlined in section 3 can offer a stepping stone framework for evolutionary research of senescence. Senescence can be viewed as a result of early vs late life trade offs¹⁰⁵. Stage structure, whether by size, social hierarchies, or division of labour¹⁰⁶, provide mechanisms to weight this trade-off in favour of late life by shifting the population’s distributions of reproductive value and class frequencies to reduce the decline of selection in later age classes. Ultimately, as Hamilton displayed (1966), and Caswell made explicit (1978) it is these two fundamental properties of a*

population that determine the strength of selection, which in turn determines the ease at which late-acting detrimental alleles may be able to invade. We must first ask how a mechanism can shift these distributions, which will then allow us understand which patterns of senescence will evolve. Hamilton's work provided fundamental insights for the evolution of senescence. Now that we have access to an unparalleled amount of data from a more diverse range of species, it is our duty as demographers, empiricists, and theoreticians, to extend the framework and once and for all solve the problem of the evolution of senescence.

L426: typo, prevalence. Also, em-dashes.

R38> Thank you for spotting, this has been corrected.

There are several typos in the reference list, please double-check it, I just scanned through it:

L476: Optimality

L480: Bacterium

L502: Williams'

L506: Antagonistic

L511: Dölling

R39> Thank you for spotting these typos, they have been corrected.

Referee 3

In this paper, the authors provide empirical data showing a lack of evidence for senescence across animals in plants. This has already been shown in Jones et al. (2013) "Diversity of ageing across the tree of life" published in Nature. But, here the authors highlight how population structure and reproductive values jointly determine the senescence pattern, and discuss how those quantities can be affected by stage structure, growth form and social interactions.

Overall, I am not convinced that the data reported here is of higher value than the one reported in Jones et al. (2013). Nonetheless, I think that the authors are making a very interesting (and often overlooked) point in showing how the reproductive value can affect the pattern of senescence, based on the mathematical derivation made by Hal Caswell (1978). I believe the authors could make their point stronger by showing those two quantities in the data they got. It is easy to calculate those quantities based on population matrices, and it would clearly show how one can dissect the factors at the origin of the senescence pattern. Additionally, the authors could be clearer about the population regulation assumption made behind each prediction regarding senescence.

I apologize for any inconvenience caused by the delay in writing my review. I hope that my comments will help the authors to make their case even stronger.

R40> We thank the reviewer for taking the time to review the manuscript and provide comments, which we have found extremely helpful. We have added reproductive value and population structure into figure 2 and the supplementary information, along with interpretation of our findings. We argue that the additional exploration of these findings, which is indeed underway as a separate manuscript in the group, is out of the scope of this review, but thank the reviewer for greatly improving our manuscript with this suggestion and validating that there is indeed value in other pieces of work we are currently pursuing! Finally, we agree with the reviewer that we were not clear about our population regulation assumptions, and hope to have clarified this throughout the manuscript.

MAJOR COMMENTS

1) The implication of reproductive value for senescence

lines 74-78: "The later age classes of such a cohort will therefore, all else being equal, contribute less to the ancestry of future generations (i.e. have lower reproductive value; Fisher 1930)" In this sentence, the authors refers to the effect of the stable age distribution on the reproductive value. I think this is wrong (but I am unsure given that I still have some hard time to grasp the concept of reproductive value). The reproductive value is independent of the proportion of individuals in each age-class. Actually, the reproductive value at age x can be expressed as the 'number of female offspring produced this year by mothers of age x or older' divided by the 'number of mothers that are at age x this year'. That is why there is this denominator term in the expression of

the reproductive value. See the bulletin ESA, "A Simple Equation for Presenting Reproductive Value to Introductory Biology and Ecology Classes", doi:10.2307/20168257; that is the neater explanation I found on the expression of the reproductive value and indeed the number of females that survive to age x is in the denominator. Therefore even if the probability to survive to an older age is by definition smaller, this low probability of surviving to old age will actually increase the reproductive value. Given that the reproductive value is one of the main focus of this paper, I would suggest the authors to define this notion very accurately. The implication of reproductive value for senescence (argued by Fisher 1930; and considered as such in this paragraph) is also misleading; that was actually the point made by Hamilton (1966). Charlesworth 2000 (in Genetics) sums up the situation at that time in a very neat way: "Hamilton (1966) noted that it is fallacious to use reproductive value as a measure of the effectiveness of selection as a function of age, and that a different measure should be used." "Neither of these formulae corresponds to reproductive value, and they have rather different implications for the relation between the age of effect of a gene and its impact on fitness" (compare his equation 2 vs. his equation 6)

Now I appreciate that it is possible to express those derivatives depending on the reproductive value (as in the equations 4 and 5, based on one of Caswell's derivations). The authors actually provide a very neat explanation in lines 245-252. Overall, the authors should be careful and extra-accurate when they describe the implication of reproductive value for senescence to avoid any confusion.

R41> We agree with this definition of individual reproductive value provided by the reviewer, and believe that the confusion in the original submission of our work was based over a difference between individual and class reproductive value. Taylor (1990) makes this important distinction. The reproductive value of an individual in a later age class may be higher because of the low probability of surviving there, but the class reproductive value of such later age classes will be lower. The class reproductive value is essentially the numerator in the selection gradient (Caswell's 1978 calculations – Eq 4/5 in our manuscript). We have clarified our definitions of reproductive value throughout the manuscript e.g. in line 214-215 in accordance with Fisher (1930).

L214-215: $v(x)$ is reproductive value⁵ (the average contribution to the ancestry of future generations)

Taylor, P.D. 1990 Allele-Frequency Change in a Class-Structured Population. *Am. Nat.* 135, 95-106

For instance, lines 119-120 are not crystal-clear: "equation 3 formulates how the force of selection acting against (hence the negative sign) an increase in age-specific mortality is proportional to reproductive value at age x ". This sounds wrong: for somebody who knows Hamilton's work but who is not familiar with Caswell's work (as I was), the expression of the reproductive value is not in equation 3.

R42 > This was a mistake which we have now corrected to "equation 3 formulates how the force of selection acting against (hence the negative sign) an increase in age-specific mortality declines from the age at first reproduction" (L97-99).

The current paper changed my mind on the implication of reproductive value for senescence. I initially believed that population structure was all that matters (I only knew of Hamilton 1966 and Charlesworth 2000). This is a strength of the paper; nonetheless, the paper remains still a bit unclear because the reproductive value is not clearly defined, and its effect on senescence is confusing in the section "Classical Evolutionary Framework of Senescence".

On the same note, I would strongly suggest the authors to show how age structure and reproductive value determine the senescence pattern (based on equation 4 and 5). It is very easy to get those quantities from the matrices they use (see doi: 10.2307/20168257 for the calculation of reproductive value). This will be very insightful to see what quantity drives the pattern of senescence, and this will considerably increase the value of the analysis shown. Additionally this will clearly illustrate how one can answer to the question "How do these distributions affect the evolutionary outcome for senescence?" (line 261) which is discussed throughout the paper.

R43> We agree with this suggestion from the reviewer, and so we have added reproductive value and population structure into figure 2 and the supplementary information, along with interpretation of our findings (L227-235).

L227-235: *Figures 2b and 2d exemplify the importance of reproductive value and population structure for explaining divergent patterns of senescence (see also Fig. S3). All species (except Pinus sylvestris; Fig. 2d) display reproductive value and stable age distributions that match their classification of actuarial senescence. Species that display positive actuarial senescence have distributions biased to earlier (st)ages, and those that do not have distributions that are more heavily skewed towards later (st)ages. Our assumption of each population sampled being at a stable state at the time of data collection provides the most intuitive reason why the patterns (including for Pinus sylvestris) may not match. It is clear from our analysis that reproductive value and population structure are key to understanding variation in patterns of senescence.*

2) Distinguishing predictions base on the assumption of exponential growth vs. density dependent growth

The authors summarize some previous results that were shown only under a specific assumption regarding the nature of the growth rate. The authors should try to be more accurate to avoid any shortcut (see the section 'Density-Dependent Population Regulation' in Moorad et al. 2019, doi:10.1016/j.tree.2019.02.006).

For instance:

lines 73-74: "There is always some non-zero probability of death, and so a cohort of individuals will decline in number over time." Actually a cohort of individuals will decline in number over time even when assuming immortality (a probability of death equal to zero) under the assumption of exponential growth; with implications for the relative strength of selection at older age vs younger age. Same in lines 90-91 that should be rephrased accordingly. Same in lines 95-96 that is true only assuming density dependence.

R44> We were initially confused with this comment but have since clarified with the reviewer, and thank the editor and reviewer for their quick response. We have decided to removed "There is always some non-zero probability of death, and so a cohort of individuals will decline in number over time" as we felt it on the whole unnecessary for our argument, especially if it may cause confusion.

lines 134-136: an age-independent change in mortality will change the selection gradient if there is density dependence (because this will cause a change in age structure), but not if an exponential growth is assumed.

R45> We thank the reviewer for spotting this and have altered our sentence to make clear our assumption of exponential growth (L109-111)

L109-111: *An age-independent change in mortality, by definition of being age-independent, will mean the overall selection gradient should still follow the same pattern over all age classes if exponential growth is assumed^{14,22}.*

MINOR COMMENTS

The author use COMPADRE Plant Matrix Database and COMADRE Animal Matrix. Could the authors be clear about whether they accounted for the "persistent problems in the construction of matrix population models" highlighted by Kendall et al. 2019 in Ecological Modelling (doi:10.1016/j.ecolmodel.2019.03.011).

R46> We can confirm so. The selection criteria we note in the methods ensures that the data we use are the most appropriate for asking questions related to senescence by following Che-Castaldo, J. *et al.*'s commentary to the above paper, of which Kendall is a co-author. We also note that the curator of both databases, and senior author of the piece by Che-Castaldo, is the senior author of this piece, and so we have been consistent with what we suggest other users to do (and don't) with regards to these databases, and what we have done here.

The authors could consider changing the title of the paper and removing "Senescence across the furthest reaches of the universe?". It means nothing to me; I guess it will confuse other readers.

R47> Changing the manuscript title was something noted by all reviewers and the editor, and so we have adjusted accordingly to "*Senescence: why and where selection gradients might not decline with age*".

lines 36-38: "Most ageing research likely stems from human desire to increase human lifespan and health span. This desire requires understanding" This does not seem very informative and relevant for the rest of the paper. Perhaps the authors could consider dropping that sentence.

R48> We have removed this sentence, and, along with guidance also from the second reviewer, have re-written the abstract.

lines 39-40: "understanding why different factors, rather than age per se, alter the sensitivity of fitness to age-related changes in mortality and reproduction". This sentence implies that previous research has focused on "understanding why age alters the sensitivity of fitness to age-related changes in mortality and reproduction", which does not sound right at all. Additionally, as argued in this manuscript, there have been extensive research on how extrinsic mortality alters the sensitivity of fitness to age-related changes in mortality. So it is not novel at all to investigate those factors that alter the strength of selection on such senescence-related mutations.

R49> Here, we were referring to the fact that senescence has been classically studied with respect to age-structured populations rather than age itself driving patterns of senescence. This wasn't clear and we have altered L33-35 of our abstract to clarify.

L33-35: *An ability to predict ageing patterns requires a firmer understanding of how and why different ecological and evolutionary factors alter the sensitivity of fitness to age-related changes in mortality and reproduction.*

lines 64-65: "The central logic common to these theories argues that the force of natural selection weakens with age". This is not entirely true given that some theories argue that antagonistic pleiotropy can favour certain genes (as written in lines 66-67: "or [selection] favours these genes"), and antagonistic pleiotropy could lead to senescence without the need for weak selection at older age; e.g., if the benefit at earlier age is massive.

R50> Although we agree with the reviewer's logic, this section of our review is describing the arguments put forward originally by Medawar, Williams & Hamilton who do indeed argue this point.

line 141: "senescence has not evolved adaptively"; this is not entirely true. The authors indeed argue that senescence can evolve via antagonistic pleiotropy. In that case, this is adaptive evolution.

R51> We have rephrased to clarify that the senescence phenotype of an allele itself is not adaptive, but because of a beneficial pleiotropic effect the allele on the whole might be considered to invade via adaptive evolution.

L118: *senescence (i) has not evolved adaptively without pleiotropy*

lines 190 and 211: "positive senescence" is not defined. I would suggest the authors to employ the term "actuarial senescence" instead, for consistently. Alternatively, the authors could employ positive senescence throughout the manuscript.

R52> We thank the reviewer for pointing this out, and have clarified the terms 'positive actuarial senescence' and 'negative actuarial senescence' in L139-142.

L139-142: *The shape metric, S , is bound between -0.5 and 0.5 , where $S > 0$ indicates that more mortality events occur at advanced ages (i.e. positive actuarial senescence), while $S < 0$ indicates low mortality late in life (i.e. negative actuarial senescence³⁵).*

line 218: should it be "long-term individually-based"?

R53> We thank the reviewer for pointing this out, and have edited accordingly.

lines 225-226: "We countered these limitations by using a thorough selection criteria". What do the authors get without using this criterion? Do they get qualitatively similar results?

R54> The selection criteria we note in the Methods (i-vii) are all required to combat issues in Kendall *et al.* 2019 that the reviewer raised above (See R48 above), as well as to ensure we only use data that makes sense to use for senescence. All data excluded from COM(P)ADRE for our research are not appropriate for asking questions related to senescence.